# Revisiting Spectral Representations in Generative Diffusion Models

**Yuehao Wang** [1]  **Peihao Wang** [1]  **Hanwen Jiang** [1,2]  **Ziyi Yang** [1]  **Qixing Huang** [1]  **Zhangyang Wang** [1]

## Abstract

Diffusion models have shown remarkable performance on diverse generation tasks. Recent work finds that imposing representation alignment on the hidden states of diffusion networks can both facilitate training convergence and enhance sampling quality, yet the mechanism driving this synergy remains insufficiently understood. In this paper, we investigate the connection between self-supervised spectral representation learning and diffusion generative models through a shared perspective on perturbation kernels. On the diffusion side, samples (e.g., images, videos) are produced by reversing a stochastic noise-injection process specified by Gaussian kernels; on the spectral representation side, spectral embeddings emerge from contrasting positive and negative relations induced by random perturbation kernels. Motivated by this, we propose a self-supervised spectral representation alignment method to facilitate diffusion model training. In addition, we clarify how joint spectral learning can benefit diffusion training from a geometric perspective. Furthermore, we find that the optimization of the spectral alignment objective is in an equivalent form of diffusion score distillation in the representation space. Building on these findings, we integrate a spectral regularizer into diffusion training objectives to improve the performance of diffusion models on multiple datasets. Experiments across images and 3D point clouds show consistent gains in generation quality.

## 1. Introduction

Diffusion models (Sohl-Dickstein et al., 2015; Song & Ermon, 2019; Ho et al., 2020; Song et al., 2021) have demonstrated strong generative capabilities across diverse domains, including images (Rombach et al., 2022; Dhariwal & Nichol, 2021), videos (Brooks et al., 2024; Bao et al., 2024), 3D shapes (Nichol et al., 2022; Zhao et al., 2025), molecules (Hoogeboom et al., 2022), etc. Their core idea is to reverse a diffusion process defined by a Gaussian perturbation kernel (Song et al., 2021). To achieve this, diffusion models learn to estimate the time-dependent score functions on *perturbed data*. Notably, this learning setup closely mirrors self-supervised representation learning, where models are also trained on data deliberately altered through perturbations or augmentations (HaoChen et al., 2021; Zbontar et al., 2021; Bardes et al., 2022; Sohn, 2016; Oord et al., 2018; Tian et al., 2020). In both cases, performance hinges on extracting useful structure from *perturbed inputs*: self-supervised methods aim to capture universal representations for downstream tasks, while diffusion models are dependent on appropriate representations to recover clean samples for the specific generation task. This parallel motivates a key question: Do diffusion models and self-supervised representation learning share a fundamental connection, and can exploiting it improve generative modeling?

Recent works have begun to explore the link between diffusion models and self-supervised representation learning (Preechakul et al., 2022; Yang et al., 2022; Abstreiter et al., 2021; Mittal et al., 2022). On the one hand, several studies reuse diffusion models as self-supervised representation learners (Chen et al., 2024; Xiang et al., 2023; Mukhopadhyay et al., 2023; Zhang et al., 2022), showing that meaningful features emerge during diffusion training and transfer well to downstream tasks (Tang et al., 2023; Park et al., 2023). On the other hand, REPA (Yu et al., 2024) takes the opposite direction, demonstrating that representation learning can in turn benefit diffusion models. By aligning the hidden states of denoising networks with clean-image embeddings from pretrained encoders such as DINOv2 (Oquab et al., 2023), REPA achieves faster convergence and stronger image generation. Nevertheless, REPA relies on representations from external foundation models, which are often unavailable for other modalities such as point clouds or graphs. Moreover, the broader intrinsic connection between diffusion and self-supervised learning remains unclear.

In this work, we conduct a pilot study on the synergy between self-supervised representation learning and diffusion-

---

[1]University of Texas at Austin, USA [2]Adobe Research, USA. Correspondence to: Qixing Huang <huangqx@cs.utexas.edu>, Zhangyang Wang <atlaswang@utexas.edu>.

*Proceedings of the 43rd International Conference on Machine Learning*, Seoul, South Korea. PMLR 306, 2026. Copyright 2026 by the author(s).

based generative modeling. Specifically, we focus on spectral representation learning (SRL) within self-supervised methods, inspired by prior works that admit multiple effective formulations built from perturbation kernels (HaoChen et al., 2021; Deng et al., 2022a; Pfau et al., 2018). Through the lens of perturbation kernels, we first review and unify the formulations of diffusion models and SRL under a shared stochastic process parameterization (Section 3.1 and Section 3.2). Given that spectral representations preserve neighborhood structure on the underlying data manifold (Deng et al., 2022a), it is plausible that incorporating spectral representation into diffusion training can inform the denoising networks of the latent, time-evolving local data geometry, thereby leading to better generative performance. Motivated by this, we propose a novel training strategy for diffusion models that regularizes the diffusion model's intermediate representations to align with the eigenfunctions of a time-varying kernel integral operator defined by a shared diffusion perturbation kernel (Section 4.2). Moreover, we establish a theoretical duality between representation learning and generative modeling (Section 4.3). In particular, we show that optimizing our spectral self-supervised objective is (in gradient) equivalent to diffusion score distillation (Poole et al., 2022) formulated via a KL divergence. This distributional alignment induces mode-seeking dynamics in representation space: embeddings are pulled toward their local data distribution and pushed away from mismatched regions, thereby facilitating the goal of generative modeling.

Experimentally, our proposed self-supervised spectral representation alignment yields consistent gains in diffusion training for image generation across four datasets with different data diversity, scales, and domains. Moreover, on point-cloud generation where pretrained encoders are unavailable, it attains strong performance over the baseline method, highlighting the method's potential to complex generative settings in which encoder pretraining is impractical.

## 2. Related Work

**Representations in Diffusion Models.** Recent work strengthens diffusion by enhancing internal representations. REPA (Yu et al., 2024) aligns denoiser features to pretrained vision encoders (e.g., DINOv2), accelerating convergence and improving sample quality. Its extensions include U-REPA for U-Nets (Tian et al., 2025), REPA-E for joint VAE training (Leng et al., 2025), VideoREPA for video (Zhang et al., 2025), and VAE-side alignment (Yao et al., 2025). REG (Wu et al., 2025) introduces a global semantic token to mitigate the lack of alignment at test time, and HASTE (Wang et al., 2025) adds holistic representation/attention alignment with an alignment-termination criterion to further speed training. However, these approaches assume access to strong foundation encoders, an assumption often violated

in resource-constrained domains (e.g., 3D shapes, proteins). Relatedly, You et al. (2023) leverages small-scale category labels, incurring additional annotation cost. Wang et al. (2024) study low-rank representations learned during diffusion model training, showing that the diffusion objective can be equivalent to a canonical subspace clustering problem.

A more relevant line of work builds on the connection between **self-supervised representation learning** and diffusion models. Early works in this direction aim to understand the internal representations of self-supervised diffusion models (Park et al., 2023; Preechakul et al., 2022; Mittal et al., 2022; Chen et al., 2024; Xiang et al., 2023; Mukhopadhyay et al., 2023; Hudson et al., 2024; Li et al., 2025). They show that hidden activations in different time steps encode semantically meaningful information that can be linearly manipulated for image editing and analysis (Park et al., 2023; Tang et al., 2023). Stoica et al. (2025) apply contrastive learning on flow trajectories, improving the uniqueness of flows. A concurrent study (Wang & He, 2025) introduces a dispersive loss that encourages internal representations of different samples to spread apart. While empirically effective, this advance offers primarily an intuitive, self-supervised rationale for improving diffusion models.

**Self-supervised representation learning.** Contrastive learning has emerged as a dominant paradigm for self-supervised visual representation learning (HaoChen et al., 2021; Wang & Isola, 2020; Tian et al., 2020). Early frameworks such as SimCLR (Chen et al., 2020) and MoCo (He et al., 2020; Chen et al., 2021) establish the importance of instance discrimination with large-scale negative sampling. Subsequent works remove the need for negatives, including BYOL (Grill et al., 2020) and SimSiam (Chen & He, 2021), showing that representation quality can emerge purely from positive-pair consistency. Other approaches reformulate contrastive learning through clustering and redundancy reduction, such as SwAV (Caron et al., 2020), Barlow Twins (Zbontar et al., 2021), and VICReg (Bardes et al., 2022). More recently, DINO (Caron et al., 2021; Oquab et al., 2023; Siméoni et al., 2025) advanced self-distillation with vision transformers, producing strong transferable features that have become standard teachers for aligning diffusion models. Collectively, these methods provide the foundation for self-supervised representation alignment in generative models.

## 3. Preliminary

### 3.1. Diffusion Models from Perturbation Kernels

In diffusion-based generative models (Ho et al., 2020; Song & Ermon, 2019; Song et al., 2021), data samples $x_0 \sim p_{\text{data}}(x_0)$ in $d$-dimensional space ($x_0 \in \mathbb{R}^d$) are first transported to a standard Gaussian distribution by gradually

perturbing the original data distribution with random Gaussian noise. Specifically, the perturbation kernel $p_{0t}(\boldsymbol{x}_t|\boldsymbol{x}_0)$ is defined as $\mathcal{N}\left(\boldsymbol{x}_t; s(t)\boldsymbol{x}_0, s(t)^2\sigma(t)^2\boldsymbol{I}\right)$, where $t$ is the timestep of the diffusion process, $s(t)$ is a scaling coefficient, and $\sigma(t)$ is the noise scale at $t$. Given this perturbation kernel, the SDE of the forward process is determined as follows:

$$\mathrm{d}\boldsymbol{x} = f(t)\boldsymbol{x}\,\mathrm{d}t + g(t)\mathrm{d}\boldsymbol{w}_t, \qquad (1)$$

where $f(t)\boldsymbol{x}$ is a drift term, $g(t) : \mathbb{R} \to \mathbb{R}$ is the diffusion coefficient of $\boldsymbol{x}$, and $\boldsymbol{w}_t$ is the standard Wiener process. The following equations describe the relations between $f(t)$, $g(t)$, $s(t)$, and $\sigma(t)$, which illustrate how the SDE can be derived from the perturbation kernel (Karras et al., 2022):

$$f(t) = \dot{s}(t)/s(t) \quad g(t) = s(t)\sqrt{2\dot{\sigma}(t)\sigma(t)}. \qquad (2)$$

Conversely, the scaling and noise scale terms in the perturbation kernel $p_{0t}$ can be rewritten with respect to $f(t)$ and $g(t)$:

$$s(t) = \exp\left(\int_0^t f(\xi)\mathrm{d}\xi\right) \quad \sigma(t) = \sqrt{\int_0^t \frac{g(\xi)^2}{s(\xi)^2}\mathrm{d}\xi}\,. \quad (3)$$

To sample the original data distribution from a randomly sampled noise, we can reverse the diffusion process. As introduced in the literature (Song et al., 2021), the reverse process of Equation 1 can be described as the SDE below:

$$\mathrm{d}\boldsymbol{x} = \left[\boldsymbol{f}(t)\boldsymbol{x} - g(t)^2\nabla_{\boldsymbol{x}}\log p_t(\boldsymbol{x})\right]\mathrm{d}t + g(t)\mathrm{d}\boldsymbol{w}_t, \quad (4)$$

where $p_t(\boldsymbol{x})$ is the perturbed data distribution evolving over the process time-dependently, and $\nabla_{\boldsymbol{x}}\log p_t(\boldsymbol{x})$ is a score function which can be estimated by training deep neural networks $\boldsymbol{s}_\phi$ to match the true scores:

$$\mathcal{L}_{\mathrm{diff}}(\phi) = \mathop{\mathbb{E}}_{t,\boldsymbol{x}_0,\boldsymbol{x}_t}\left[\omega_t\left\|\boldsymbol{s}_\phi(\boldsymbol{x}_t,t) - \nabla_{\boldsymbol{x}_t}\log p_{0t}(\boldsymbol{x}_t|\boldsymbol{x}_0)\right\|_2^2\right]$$
$$(5)$$
$$= \mathop{\mathbb{E}}_{\substack{t,\ \boldsymbol{x}_0\sim p_{\mathrm{data}}\\\boldsymbol{x}_t\sim p_{0t}(\boldsymbol{x}_t|\boldsymbol{x}_0)}}\left[\omega_t\left\|\boldsymbol{s}_\phi(\boldsymbol{x}_t,t) + \frac{\boldsymbol{x}_t - s(t)\boldsymbol{x}_0}{s(t)^2\sigma(t)^2}\right\|_2^2\right], \quad (6)$$

where $\omega_t$ is a time-dependent re-weighting of score-matching losses across different $t$. Formulating diffusion processes with perturbation kernels facilitates score matching in the two aspects: 1) Given $\boldsymbol{x}_0$ and $\boldsymbol{x}_t$, the true scores have analytic expressions. 2) The perturbation kernel $p_{0t}$ allows for a "simulation-free" forward process, i.e., one can sample $\boldsymbol{x}_t = s(t)\boldsymbol{x}_0 + s(t)\sigma(t)\epsilon$ without numerically simulating the SDE in Equation 1. Moreover, flow-based diffusion models (Liu et al., 2022) can be defined by perturbation kernels as well (see Appendix A for the derivation).

## 3.2. Spectral Representation from Perturbation Kernels

In this section, we will revisit a family of self-supervised learning approach that restores data representations in the spectral domain of kernels, a.k.a spectral representation learning (SRL).

**Spectral Contrastive Learning.** SCL (HaoChen et al., 2021) reframes self-supervised representation learning as a spectral decomposition of a population-level augmentation graph. Unlike traditional methods that rely on the assumption that positive pairs are conditionally independent given their labels, SCL constructs graphs using data points, where edges connect different augmentations of the same underlying data point. By minimizing the contrastive objective (Equation 8), the neural network is theoretically guaranteed to perform spectral decomposition on the population augmentation graph.

$$\mathcal{L}_{SC} = -2\mathbb{E}_{\boldsymbol{x},\boldsymbol{x}^+}[\psi(\boldsymbol{x})^\top\psi(\boldsymbol{x}^+)] \qquad (7)$$
$$+ \mathbb{E}_{\boldsymbol{x},\boldsymbol{x}'}[(\psi(\boldsymbol{x})^\top\psi(\boldsymbol{x}'))^2], \qquad (8)$$

where $\boldsymbol{x}$ and $\boldsymbol{x}^+$ are two views sampled from a perturbation kernel $p(\cdot|\bar{\boldsymbol{x}})$ (augmentation) on the same data point $\bar{\boldsymbol{x}} \sim p_{\mathrm{data}}$ (clean data distribution), $\boldsymbol{x}$ and $\boldsymbol{x}'$ are two samples augmented from independent data points, and $\psi$ is a neural network. Johnson et al. (2022) further find that this learning objective is a special case of kernel learning. Specifically, the target kernel learned by the networks can be written as:

$$\kappa(\boldsymbol{x},\boldsymbol{x}') = \frac{p(\boldsymbol{x},\boldsymbol{x}')}{p(\boldsymbol{x})p(\boldsymbol{x}')}, \qquad (9)$$
$$p(\boldsymbol{x},\boldsymbol{x}') = \mathbb{E}_{\bar{\boldsymbol{x}}\sim p_{\mathrm{data}}}[p(\boldsymbol{x}|\bar{\boldsymbol{x}})p(\boldsymbol{x}'|\bar{\boldsymbol{x}})]. \qquad (10)$$

Through this perspective, the objective of SCL is to learn the kernel principal components.

**Neural Eigenmap.** Deng et al. (2022a) propose to formalize spectral representation learning by solving ordered eigenfunctions of the kernel integral operator. Given the kernel $\kappa(\boldsymbol{x},\boldsymbol{x}')$ in Equation 10, the corresponding kernel integral operator is defined in the following way:

$$(\mathcal{T}_\kappa h)(\boldsymbol{x}) = \int \kappa(\boldsymbol{x},\boldsymbol{x}')f(\boldsymbol{x}')p(\boldsymbol{x}')d\boldsymbol{x}', \qquad (11)$$

where $f \in L^2(\mathcal{X}, p)$, i.e., $f$ is a square-integrable function w.r.t $p$. $\mathcal{X}$ is a support, and $p$ is a probability distribution defined over the support. Intuitively, this operator can be understood as the continuous-domain analogue of matrix multiplication. Similar to the result in Johnson et al. (2022), Neural Eigenmap trains a neural network to approximate the principal eigenfunctions of the kernel integral operator. Then, the spectral representation learning objective becomes solving the eigenvalue problem. Following NeuralEF (Deng

et al., 2022b), Neural Eigenmap reformulates the eigenfunction problem of $\mathcal{T}_\kappa \psi^j = \mu \psi^j$ into an optimization problem:

$$\max_{\psi_j} R_{j,j} - \alpha \sum_{i=1}^{j-1} R_{i,j}^2, \quad \text{for } j = 1, .., K, \quad (12)$$

$$R = \mathbb{E}_{p(\boldsymbol{x},\boldsymbol{x}')} \left[ \psi(\boldsymbol{x})\psi(\boldsymbol{x}')^\top \right] \approx \frac{1}{B} \sum_{b=1}^{B} \psi(\boldsymbol{x}_b)\psi(\boldsymbol{x}_b')^\top, \quad (13)$$

where $K$ is the number of eigenfunctions, $\psi(\boldsymbol{x}) = \left[ \psi^1(\boldsymbol{x}), ..., \psi^K(\boldsymbol{x}) \right] \in \mathbb{R}^K$ denotes the vector comprising the first $K$ eigenfunctions evaluated at $\boldsymbol{x}$, $B$ is the number of data samples, $\boldsymbol{x}_b$ and $\boldsymbol{x}_b'$ are independently sampled from the perturbation kernel $p(\boldsymbol{x}|\bar{\boldsymbol{x}}_b)$ conducted on the same clean data $\bar{\boldsymbol{x}}_b$. We can parameterize $\psi$ by a neural network, and the network parameters $\theta$ can be optimized through the following loss function, which bears a strong resemblance to other contrastive representation learning objectives (Li et al., 2022; Zbontar et al., 2021):

$$\mathcal{L}_{ef}(\theta) = - \sum_{j=1}^{K} \left( \psi_\theta(\boldsymbol{X}_B)\psi_\theta(\boldsymbol{X}_B')^\top \right)_{j,j} \quad (14)$$

$$+ \alpha \sum_{j=1}^{K} \sum_{i=1}^{j-1} \left( \text{sg}(\psi_\theta(\boldsymbol{X}_B))\psi_\theta(\boldsymbol{X}_B')^\top \right)_{i,j}^2, \quad (15)$$

where $\text{sg}(\cdot)$ denotes stop-gradient operator that converts its argument as an constant with zero derivative, $\alpha$ is the coefficient weighting the regularization applied to the upper-triangular elements, $\boldsymbol{X}_B = [\boldsymbol{x}_1, ..., \boldsymbol{x}_B]$, $\boldsymbol{X}_B' = [\boldsymbol{x}_1', ..., \boldsymbol{x}_B']$ are batched input data, $\boldsymbol{x}_b$ and $\boldsymbol{x}_b'$ are perturbed from the same clean data $\bar{\boldsymbol{x}}_b$ for $b = 1, ..., B$, and $B$ is the batch size for mini-batch training. Thereby $\psi_\theta(\boldsymbol{X}_B)$ is a $K \times B$ matrix with the element at $j$-th row, $b$-th column representing the $j$-th eigenfunction evaluated at the $b$-th data sample in the training batch.

The perturbation kernels $p(\boldsymbol{x}|\bar{\boldsymbol{x}})$ used for SRL are usually designed as composed data augmentations. For instance, for representation learning on images, $p(\boldsymbol{x}|\bar{\boldsymbol{x}})$ can be a composition of image manipulations, such as color jittering, random flip, Gaussian blur, etc.

# 4. Bridging Spectral Representations and Diffusion Models

We have reviewed diffusion models and spectral representations through the lens of perturbation kernels. Motivated by their shared principle of learning from perturbed data, we further develop their connections and propose a spectral representation alignment approach.

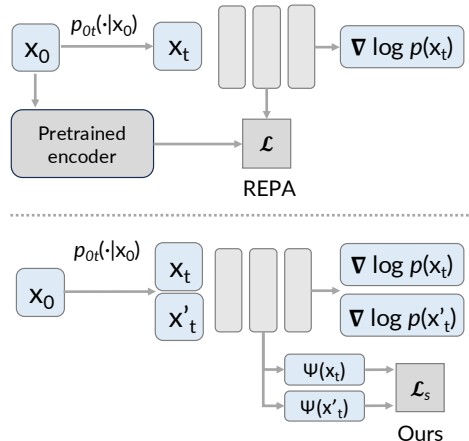

*Figure 1.* Pipeline comparison between the external encoder-based REPA method (Yu et al., 2024) (top) and our self-supervised alignment method (bottom). Given a clean sample, we independently draw two noisy views, reusing the perturbation kernel adopted by the diffusion model. The model then learns time-dependent spectral representations via self-supervised signals that align noisy views while implicitly separating unrelated samples.

## 4.1. Spectral Geometry in Diffusion Process

To study the synergy of SRL and diffusion models, we adopt the same perturbation kernel in diffusion models, i.e, $p_{0t}(\boldsymbol{x}_t|\boldsymbol{x}_0)$. Therefore, once the SDE of a diffusion process is given, a time-dependent perturbation kernel $\kappa_t$ is also determined for SRL:

$$\kappa_t(\boldsymbol{x}, \boldsymbol{x}') = \frac{p_t(\boldsymbol{x}, \boldsymbol{x}')}{p_t(\boldsymbol{x})p_t(\boldsymbol{x}')} \quad (16)$$

$$p_t(\boldsymbol{x}, \boldsymbol{x}') = \mathbb{E}_{\boldsymbol{x}_0 \sim p_{\text{data}}}[p_{0t}(\boldsymbol{x}_t|\boldsymbol{x}_0)p_{0t}(\boldsymbol{x}_t'|\boldsymbol{x}_0)]. \quad (17)$$

Using this kernel, we can construct its time-varying kernel integral operator $\mathcal{K}_t$:

$$(\mathcal{K}_t h)(\boldsymbol{x}) = \int \kappa_t(\boldsymbol{x}, \boldsymbol{x}')h(\boldsymbol{x}')p_t(\boldsymbol{x}')d\boldsymbol{x}'. \quad (18)$$

Since this operator is time-varying, its eigenfunctions also need to be formulated in a time-dependent manner: $\mathcal{K}_t \psi_\theta^j(\boldsymbol{x}_t, t) = \mu_t \psi_\theta^j(\boldsymbol{x}_t, t)$. The time-dependent eigenfunctions preserve the local geometry of data points on a latent, time-evolving manifold. This follows the classical spectral paradigm: in algorithms such as spectral clustering (Ng et al., 2001; Shi & Malik, 2000) and diffusion maps (Coifman & Lafon, 2006; Coifman et al., 2005; Nadler et al., 2005), eigenspace embeddings of constructed kernel operators yield coordinates that respect neighborhood structure and facilitate unsupervised clustering. In our setting, the kernel operator $\mathcal{K}_t$ varies with time via the SDE-defined perturbation (Marshall & Hirn, 2018), and the embeddings $\psi_\theta(\mathbf{x}_t, t)$ track the local connectivity as it evolves following the diffusion process. Inspired by (Coifman & Lafon, 2006; Nadler et al., 2006; Coifman et al., 2008), we formalize a

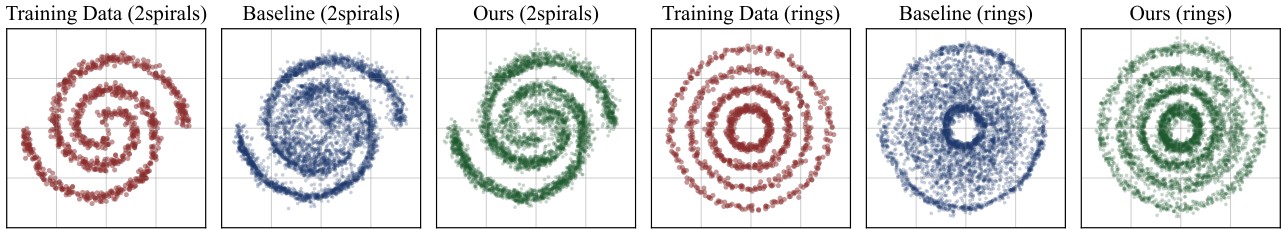

*Figure 2.* **Results on synthetic 2D data distributions.** Our method produces a cleaner, more compact sample distribution than the baseline, with fewer outliers.

diffusion distance induced by $\kappa_t(\boldsymbol{x}, \boldsymbol{x}')$ to characterize the time-varying local connectivity of the data manifold, which can be approximated by the eigenfunctions of $\mathcal{K}_t$ (Proposition 4.2). Unlike existing approaches in the literature, the diffusion distance in our case is derived from the joint probability between two points rather than relying on a predefined affinity kernel.

**Definition 4.1.** Given the kernel $\kappa_t(\boldsymbol{x}, \boldsymbol{x}')$, diffusion distance can be defined as follows.

$$D^2_{\kappa_t}(\boldsymbol{x}, \boldsymbol{x}') = \int \left[ \kappa_t(\boldsymbol{x}, \boldsymbol{y}) - \kappa_t(\boldsymbol{x}', \boldsymbol{y}) \right]^2 p_t(\boldsymbol{y}) \, d\boldsymbol{y} \quad (19)$$

**Proposition 4.2.** *The diffusion distance $D^2_{\kappa_t}(\boldsymbol{x}, \boldsymbol{x}')$ admits an expansion in the eigenspace of the associated kernel integral operator $\mathcal{K}_t$.*

$$D^2_{\kappa_t}(\boldsymbol{x}, \boldsymbol{x}') = \sum_{l=0}^{\infty} \mu^2_{t,l} \left[ \psi^l(\boldsymbol{x}, t) - \psi^l(\boldsymbol{x}', t) \right]^2, \quad (20)$$

*where $\mu_{t,l}$ is the eigenvalue of $\psi^l(\boldsymbol{x}, t)$.*

### 4.2. Spectral Representation Alignment

Recent work REPA (Yu et al., 2024) shows that representation alignment can result in better generation performance. This motivates us to incorporate SRL as a regularizer within diffusion training. Unlike REPA, which aligns the hidden states of diffusion transformers with external teacher signals, our adopted SRL is a fully self-supervised objective. This eliminates dependency on large-scale pretrained encoders, such as DINO and CLIP, which are usually computationally costly and even unavailable in data-constrained settings.

To establish compatibility between the two objectives, we first recast spectral learning in terms of the diffusion perturbation kernel $p_{0t}(\boldsymbol{x}_t|\boldsymbol{x}_0) = \mathcal{N}(\boldsymbol{x}_t; (1-t)\boldsymbol{x}_0, t^2\boldsymbol{I})$ (the one used in rectified flow), where $\boldsymbol{x}_0$ is a clean data sampled from $p_{\text{data}}$. Note that our subsequent analysis is insensitive to the specific parameterization of the perturbation kernel; the particular choices of $s(t)$ and $\sigma(t)$ for $p_{\text{data}}$ will not affect our following discussion.

Plugging $\kappa_t$ into Neural Eigenmap, we can solve the eigen-

function problem using the following spectral loss:

$$\mathcal{L}_s(\theta) = \mathbb{E}_{\substack{t, \; \boldsymbol{x}_0 \sim p_{\text{data}} \\ \boldsymbol{x}_t, \boldsymbol{x}'_t \sim p_{0t}(\boldsymbol{x}_t|\boldsymbol{x}_0)}} \left[ - \operatorname{Tr}\left( \psi_\theta(\boldsymbol{x}_t, t) \psi_\theta(\boldsymbol{x}'_t, t)^\top \right) \right.$$
$$\left. + \alpha \sum_{j=1}^{K} \sum_{i=1}^{j-1} \left( \operatorname{sg}\left( \psi_\theta(\boldsymbol{x}_t, t) \right) \psi_\theta(\boldsymbol{x}'_t, t)^\top \right)^2_{i,j} \right],$$
$$(21)$$

where $t \in (0, 1]$ is a randomly sampled time step, $\boldsymbol{x}_t$ and $\boldsymbol{x}'_t$ are two i.i.d perturbed views of the same clean data samples $\boldsymbol{x}_0$, and the time-conditioned neural network $\psi_\theta(\boldsymbol{x}_t, t)$ parameterizes the eigenfunctions of $\mathcal{K}_t$. Comparing $\mathcal{L}_s$ and $\mathcal{L}_{\text{diff}}$ in Equation 6, both involve sampling random time steps $t$ and perturbed data $\boldsymbol{x}_t \sim p_{0t}(\boldsymbol{x}_t|\boldsymbol{x}_0)$, whereas Equation 21 additionally requires an independently sampled $\boldsymbol{x}'_t$. This permits a practical implementation that jointly optimizes the diffusion and spectral objectives while reusing the same perturbed input, leading to our final training objective:

$$\mathcal{L}(\theta, \phi) = \mathcal{L}_{\text{diff}}(\phi) + \lambda \mathcal{L}_s(\theta), \quad (22)$$

where $\theta$ denotes parameters of the spectral learner $\psi_\theta$, $\phi$ is a set of parameters of diffusion networks, and $\lambda$ is the coefficient controlling the strength of the spectral regularization.

As discussed in Section 4.1, the SRL term in the objective yields multi-scale representations that reflect the intrinsic local connectivity at each $t$: for small $t$, data remain well separated, so only nearby points have small embedding distances; as $t$ increases and noise dominates, eigenspace distances progressively collapse and become less discriminative. Therefore, spectral alignment of the hidden states enables the diffusion denoiser to characterize the time-varying geometric priors inherent in the data manifold. To empirically validate this , Section 5.1 evaluates our approach on synthetic 2D distributions of special geometric patterns.

**Implementation Details.** We follow the implementation of representation alignment in REPA. The diffusion denoiser and spectral representation learner share the same backbone. The output of a specified intermediate layer will be probed to a projection head for alignment. The projection head is a two-layer MLP. We condition it on the timestep, identical to the time modulation in (Peebles & Xie, 2023). We apply L2-BN at the final layer to enforce a normalization constraint

on the estimated eigenfunctions (Deng et al., 2022b). To stabilize training, we also normalize each output embedding to bound its magnitude. Figure 1 illustrates the comparison between REPA and our method.

### 4.3. Spectral Representation Learning as Diffusion Score Distillation

We further look into the self-supervised learning objective in Equation 21. Unlike sample-contrastive methods, Equation 21 does not explicitly construct negative examples. Consequently, the spectral regularizer belongs to the dimension-contrastive family in Garrido et al. (2022), which is provably dual to sample-contrastive learning with positives and negatives. From this viewpoint, for a given perturbed sample as an anchor, instances perturbed from different clean examples can be interpreted as negatives, whereas instances perturbed from the same clean example play the role of positives. Interestingly, in its dual (sample-contrastive) form, our spectral regularizer admits a reformulation as diffusion score distillation (Poole et al., 2022).

**Proposition 4.3.** *Minimizing the self-supervised learning objective in Equation 21 via a gradient-based optimizer is equivalent to minimizing the KL divergence $D_{KL}(p_t^{\psi_\theta}(\boldsymbol{x}_t) \parallel p_+)$, as the following identity shows:*

$$\frac{\partial \mathcal{L}_s}{\partial \theta} = \mathbb{E}_{\boldsymbol{x} \sim p_t} \left[ (\nabla_\theta \psi_\theta(\boldsymbol{x}, t))^\top \nabla_{\psi_\theta(\boldsymbol{x}, t)} \mathcal{L}_s \right] \quad (23)$$

$$\equiv \nabla_\theta D_{KL}(p_t^{\psi_\theta} \parallel p_+) \quad (24)$$

*where $\nabla_{\boldsymbol{x}_t} \log p_t^{\psi_\theta}$ is equal to the closed-form diffusion scores (Scarvelis et al., 2023) evaluated over negative samples, and the target score $\nabla_{\boldsymbol{x}_t} \log p_+$ matches the closed-form diffusion scores evaluated over positive samples.*

Complete steps to show the above proposition are provided in Appendix C. Intuitively, this KL term measures, at the anchor representation $\psi_\theta(\mathbf{x}_t)$, the discrepancy between a distribution of negative samples and a distribution of positive samples. Since our spectral regularizer applies a stop-gradient to the negatives, minimizing $D_{KL}(p_t^{\psi_\theta} \parallel p_+)$ updates $\theta$ so that the anchor $\psi_\theta(\mathbf{x}_t)$ moves to reconcile the score fields of the positive and negative distributions. The resulting dynamics are mode-seeking in representation space, tightening clusters of similar samples while pushing dissimilar ones apart.

## 5. Experiments

We evaluate our approach across 2D pattern fitting (Section 5.1), image (Section 5.2) and point cloud (Section 5.3) generation tasks. These experiments are specifically designed to demonstrate the efficacy of our approach in **domains lacking external pretrained encoders**, such as 3D point clouds and low-resolution images. In

### 5.1. Synthetic Distributions

We first validate the effectiveness of our approach on synthetic 2D distributions. We consider the "2-spirals" and "rings" patterns, which exhibit intricate geometric structures. For each 2D pattern, 1K points are randomly drawn as the training set. We then train a simple MLP using either the vanilla diffusion loss or the diffusion loss augmented with our spectral regularizer. The models are trained for 5K epochs on 2-spirals and 10K epochs on rings, respectively. In Figure 2, we visualize 3K samples generated by the baseline model and our spectral alignment model. Our method yields cleaner, more compact samples with markedly fewer out-of-distribution points. On the "2-spirals" pattern, it recovers the fine spiral geometry that the baseline misses. For the "rings" pattern, our model achieves a tighter fit to the underlying shape of concentric circles. While the baseline samples show noticeable dispersion. Figure 7 presents additional results on other 2D data distributions. These results illustrate that our proposed method can capture the underlying data geometry prior more efficiently.

### 5.2. Image Generation

**Dataset.** We test our method on CIFAR10 (Krizhevsky et al., 2009), CelebA (Liu et al., 2015), FFHQ (Karras et al., 2019), ImageNet (Deng et al., 2009) datasets, which are standard datasets used for training image generation with different data diversity, domain, and scale. For CIFAR10 and CelebA datasets, we resize images into $32 \times 32$ resolution. While for FFHQ, images are resized to $64 \times 64$. For ImageNet, we resize images to two different resolutions: $64 \times 64$ and $256 \times 256$. For ImageNet $256 \times 256$ experiments, each image is further encoded to $32 \times 32 \times 4$ latents using Stable Diffusion VAE (Rombach et al., 2022), and latent diffusion models are trained on those encoded latents. For other image generation tasks, we conduct diffusion model training on pixel space.

**Training details.** We use DiT (Peebles & Xie, 2023) as the base model and employ the parameterization and training objective of rectified flow (Liu et al., 2022). More details are provided in Appendix D.2.

**Evaluation protocol and baselines.** We evaluate generation quality using Fréchet Inception Distance (FID) as the primary metric, complemented by sFID, Inception Score (IS), and the precision/recall pair as secondary measures. All the reported metrics are measured on EMA checkpoints. For pixel-space diffusion, we compare against a vanilla DiT baseline trained under the same setting with ours except no use of our proposed representation learning loss. To further understand the effectiveness of our proposed method, for latent diffusion, we also compare against REPA (Yu et al., 2024), a leading representation-alignment method that lever-

| Dataset | Model | Metric | | | | |
|---------|-------|--------|--------|-------|-------------|----------|
| | | FID ($\downarrow$) | sFID ($\downarrow$) | IS ($\uparrow$) | Precision ($\uparrow$) | Recall ($\uparrow$) |
| ImageNet (res. = 64) | DiT-L/4 baseline | 9.441 | 7.653 | **102.069** | **0.871** | 0.393 |
| | Ours (DiT-L/4) | **7.994** | **7.372** | 78.366 | 0.858 | **0.397** |
| ImageNet (res. = 256, latent) | DiT-XL/2 baseline | 2.508 | 5.630 | 247.891 | 0.822 | 0.566 |
| | REPA (DiT-XL/2) | **1.745** | **5.459** | **296.726** | 0.807 | **0.615** |
| | Ours (DiT-XL/2) | 2.298 | 5.510 | 257.741 | **0.824** | 0.570 |
| CIFAR10 (res. = 32) | DiT-S/2 baseline | 11.588 | 10.680 | 9.042 | 0.719 | 0.384 |
| | Ours (DiT-S/2) | **8.742** | **6.836** | **9.174** | **0.735** | **0.405** |
| CelebA (res. = 32) | DiT-S/2 baseline | 28.806 | 20.569 | **3.431** | 0.685 | 0.453 |
| | Ours (DiT-S/2) | **25.678** | **20.061** | 3.388 | **0.702** | **0.472** |
| FFHQ (res. = 64, uncond.) | DiT-S/2 baseline | 13.766 | 21.982 | 2.997 | 0.731 | 0.331 |
| | Ours (DiT-S/2) | **13.074** | **21.915** | **2.998** | **0.737** | **0.340** |

*Table 1.* **Evaluation of image generation across four datasets**, with image resolutions and model sizes adapted accordingly. We report FID as the primary metric, and sFID, Inception Scores, Precision/Recall as secondary metrics.

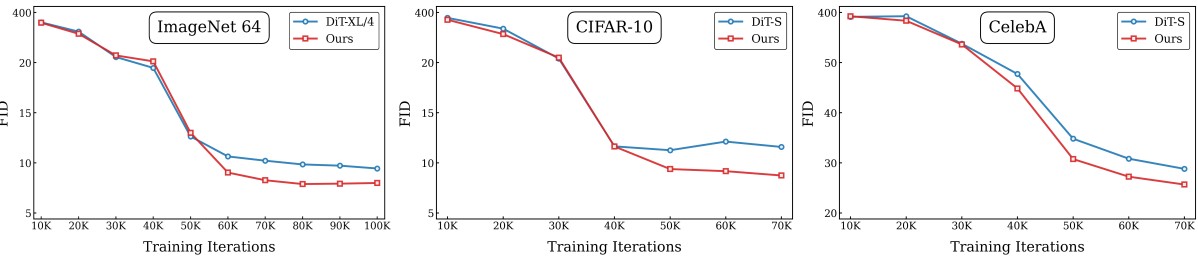

*Figure 3.* **Visualization of Training Progress.** We plot FID against training iterations for three datasets. These results suggest that our representation learning strategy sustains effective optimization and mitigates the mid-training stagnation observed in the baseline.

ages encoders pretrained on large-scale external data, which serves as the upper bound of performance. We employ Euler ODE for pixel-space generation and SDE Euler-Maruyama sampler for latent-space generation. For conditional generation, we set CFG=2.

**Results.** Table 1 compares our method with matched diffusion baselines under the same backbone and training setting. Across the evaluated image-generation benchmarks, our spectral regularization consistently improves FID over the corresponding baseline. Specifically, our method reduces FID by 1.5 on ImageNet-64 with DiT-L/4, 0.2 on ImageNet-256 with DiT-XL/2, 2.8 on CIFAR-10, 3.1 on CelebA, and 0.7 on FFHQ, corresponding to relative improvements of 15%, 8%, 25%, 11%, and 5%, respectively. These results suggest that the proposed self-supervised spectral objective can provide a stable improvement over the underlying diffusion backbone across different resolutions, model scales, and both pixel- and latent-space generation settings. For latent-space ImageNet-256 generation, REPA achieves the best overall performance, while our method improves over the baseline without relying on an external pretrained encoder. We further evaluate performance throughout training.

As shown in Figure 3, our method outperforms the baseline in the later stages of training, suggesting that the improvement is not limited to the earliest optimization phase. Additional results are provided in Appendix D.2.

### 5.3. Point Cloud Generation

**Dataset.** Following prior work (Yang et al., 2019; Mo et al., 2023), we use the ShapeNet (Chang et al., 2015) Chair, Airplane, and Car categories with the same preprocessing and data split as Yang et al. (2019). We sample 2,048 points for each shape instance.

**Training details.** For each subset, we use DiT-3D model (Mo et al., 2023) as the base model, which employs 3D window attention in transformer blocks. As the dataset of 3D shapes is relatively small, we use the S/4 configuration (33M parameters, patch size 4). We train the models on each shape category for 10k iterations. We use the same batch-size scheme as in the image-generation experiments.

**Evaluation protocol and baseline.** We follow the setup in DiT-3D to evaluate the generated samples with 1-nearest

| Dataset | Iteration | Model | 1-NNA (↓) | | COV (↑) | |
|---|---|---|---|---|---|---|
| | | | CD | EMD | CD | EMD |
| Chair | 5K | DiT 3D-S/4 baseline | 0.850 | 0.875 | 0.295 | 0.221 |
| | | Ours (DiT 3D-S/4) | **0.583** | **0.627** | **0.488** | **0.493** |
| | 10K | DiT 3D-S/4 baseline | 0.565 | 0.545 | 0.504 | 0.511 |
| | | Ours (DiT 3D-S/4) | **0.520** | **0.527** | **0.517** | **0.543** |
| Airplane | 5K | DiT 3D-S/4 baseline | 0.714 | 0.668 | 0.522 | 0.519 |
| | | Ours (DiT 3D-S/4) | **0.601** | **0.556** | **0.561** | **0.523** |
| | 10K | DiT 3D-S/4 baseline | 0.852 | 0.785 | 0.397 | 0.389 |
| | | Ours (DiT 3D-S/4) | **0.607** | **0.562** | **0.570** | **0.600** |
| Car | 5K | DiT 3D-S/4 baseline | 0.788 | 0.738 | 0.378 | 0.482 |
| | | Ours (DiT 3D-S/4) | **0.605** | **0.586** | **0.458** | **0.549** |
| | 10K | DiT 3D-S/4 baseline | 0.730 | 0.682 | 0.427 | 0.427 |
| | | Ours (DiT 3D-S/4) | **0.582** | **0.500** | **0.505** | **0.573** |

*Table 2.* **Evaluation of 3D point cloud generation** on three subsets of ShapeNet objects. We include 1-NNA and COV computed by either using chamfer distance (CD) or earth mover's distance (EMD) as the criterion for shape retrieval.

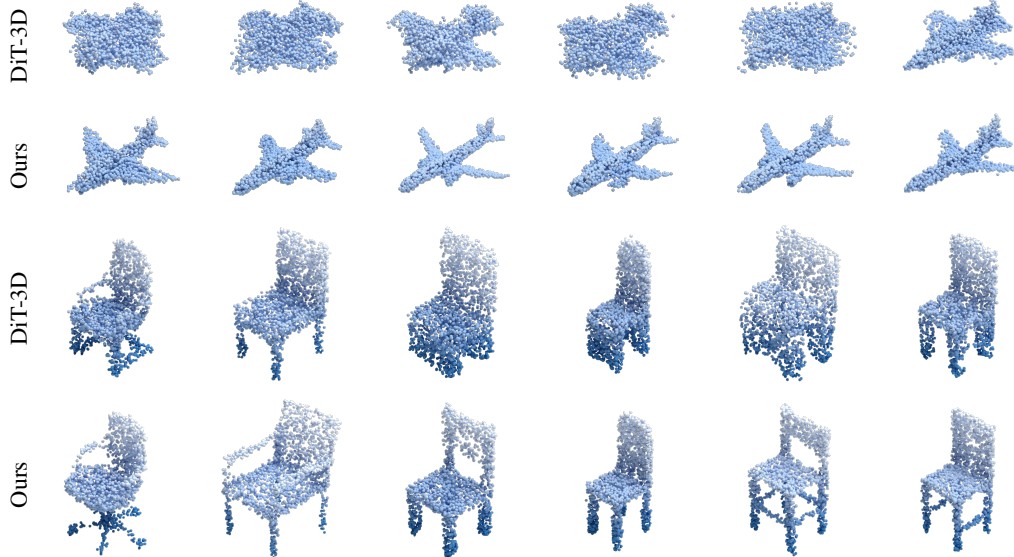

*Figure 4.* **Visualization of point cloud generation results.** We include generated samples on airplanes (top two rows, generated when model trained with 3K iterations) and chairs (bottom two rows, generated when model trained with 5K iterations).

neighbor accuracy (1-NNA) and generated sample coverage (COV). For each metric, Chamfer Distance (CD) and Earth Mover's Distance (EMD) are used to measure the distance between 3D shapes.

**Qualitative results.** Figure 4 shows comparisons between generated point clouds of our method and the baseline in "Chair" and "Airplane" categories. Our method demonstrates significantly faster convergence compared to DiT-3D. At an early training stage (3K iterations for airplanes and 5K iterations for chairs), the generations from DiT-3D remain noisy and fragmented, producing messy point distributions without clear geometric structure. In contrast, our approach already produces compact and coherent point clouds that exhibit well-defined shapes with fine-grained details.

**Quantitative results.** Table 2 presents the point cloud generation evaluation results, where our method consistently demonstrates both faster convergence and superior final performance compared to the DiT 3D-S/4 baseline. Notably, after only 5K iterations, our approach already achieves substantial improvements across all datasets. For instance, on the Chair dataset, the 1-NNA (CD/EMD) drops from 0.850/0.875 to 0.583/0.627 (31% and 28% relative improvement, respectively), while the COV (CD/EMD) rises from 0.295/0.221 to 0.488/0.493 (65% and 123% relative improvement, respectively). Similar trends are observed for Airplane and Car, where our model attains a lower 1-NNA and a higher COV at the early stage of training. With longer training, our method further improves upon these gains, achieving the best overall results across all metrics.

## 5.4. Limitations

Our empirical validation is limited to relatively small-scale datasets and model sizes compared with state-of-the-art diffusion models. Therefore, our results mainly demonstrate improvements over matched diffusion backbones under the same training and sampling settings. Our method does not outperform external-teacher approaches such as REPA, which remain stronger in large-scale latent image generation by leveraging pretrained encoders. In addition, our method introduces extra training overhead because it requires an additional perturbed view and a projection head for representation alignment. Under our implementation and experimental setup, the training time increases by approximately 10% for ImageNet-64 pixel-space diffusion, 70% for ImageNet-256 latent-space diffusion due to the larger hidden-state dimensionality, and 20% for 3D diffusion on Airplane point clouds.

## 6. Conclusion

In this work, we investigate the connection between self-supervised spectral representation learning and diffusion models through the shared lens of perturbation kernels. Leveraging this alignment, we introduce a spectral representation alignment approach to diffusion models, offer a geometric interpretation of why joint spectral learning benefits diffusion training, and establish its equivalence to diffusion score distillation in representation space. Integrating the resulting spectral regularizer into standard diffusion objectives yields consistent gains on image and 3D point cloud generation. These findings suggest a practical, principled path for further exploring the synergy between diffusion modeling and representation learning.

## Impact Statement

This work aims to improve the performance of generative models, particularly in settings where training data are limited or exhibit specialized structure. By enabling more effective training of diffusion-based generative models under such constraints, the proposed approach has the potential to reduce training time and computational cost, thereby contributing to improved energy efficiency.

## Acknowledgements

PW is in part supported by Google PhD Fellowship in Machine Learning and ML Foundations. QH is supported by NSF 2047677, 2413161, 2504906, 2515626, and Gifts from Adobe and Google. ZW is supported in part by NSF Awards 2145346 (CAREER), 2523383 (DMS), and the NSF AI Institute for Foundations of Machine Learning (IFML). Authors acknowledge the computing support on the Vista GPU Cluster through the Center for Generative AI (CGAI) and TACC at UT Austin.

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

## A. Perturbation Kernels of Rectified Flow

We show how to derive the forward SDE of rectified flow (Liu et al., 2022) from its perturbation kernel. Note that, the forward process in the original rectified flow is originally defined as $\boldsymbol{x}_t = (1-t)\boldsymbol{x}_0 + t\boldsymbol{\epsilon}$, $\boldsymbol{x}_0 \sim p_{\text{data}}$, $\boldsymbol{\epsilon} \sim \mathcal{N}(0, \boldsymbol{I})$. It appears this forward process is a linear interpolation between random noise and clean data samples, rather than in the form of SDE. In fact, it can be rewritten as an SDE using the perturbation kernel defined by the interpolation: $p_{0t}(\boldsymbol{x}_t|\boldsymbol{x}_0) = \mathcal{N}(\boldsymbol{x}_t; (1-t)\boldsymbol{x}_0, t^2\boldsymbol{I})$. Then, $s(t) = 1 - t$, $\sigma(t) = \frac{t}{1-t}$. By Equation 2, $f(t) = -\frac{1}{1-t}$, $g(t) = \sqrt{\frac{2t}{1-t}}$. Then, we can write down the forward SDE as:

$$\mathrm{d}\boldsymbol{x} = -\frac{1}{1-t}\,\boldsymbol{x}\,\mathrm{d}t + \sqrt{\frac{2t}{1-t}}\mathrm{d}\boldsymbol{w}_t. \tag{25}$$

The corresponding reverse SDE is:

$$\mathrm{d}\boldsymbol{x} = \left[-\frac{1}{1-t}\,\boldsymbol{x} - \frac{2t}{1-t}\nabla_{\boldsymbol{x}}\log p_t(\boldsymbol{x})\right]\mathrm{d}t + \sqrt{\frac{2t}{1-t}}\mathrm{d}\boldsymbol{w}_t. \tag{26}$$

This SDE can be further converted into an ODE that preserves the marginal distribution $p_t(\boldsymbol{x})$:

$$\mathrm{d}\boldsymbol{x} = \underbrace{-\frac{1}{1-t}\left[\boldsymbol{x} + t\nabla_{\boldsymbol{x}}\log p_t(\boldsymbol{x})\right]\mathrm{d}t}_{\text{velocity field: } \boldsymbol{v}_t(\boldsymbol{x})}, \tag{27}$$

which yields the velocity field directly adopted in the original rectified flow approach. This relation between the score function and the velocity field in rectified flow is also shown in CFDM (Scarvelis et al., 2023).

## B. Geometric Interpretation of Representations in Eigenspace

In this section, we aim to give an interpretation of the representation learned from the diffusion process. We first define the time-dependent perturbation kernel induced by the diffusion process. Let $q_t(\boldsymbol{x} \mid \boldsymbol{x}_0) := p_{0t}(\boldsymbol{x} \mid \boldsymbol{x}_0)$. Then, we have:

$$p_t(\boldsymbol{x}) = \int q_t(\boldsymbol{x} \mid \boldsymbol{x}_0)p_{\text{data}}(\boldsymbol{x}_0)d\boldsymbol{x}_0. \tag{28}$$

We define

$$p_t(\boldsymbol{x}, \boldsymbol{x}') = \int q_t(\boldsymbol{x} \mid \boldsymbol{x}_0)q_t(\boldsymbol{x}' \mid \boldsymbol{x}_0)p_{\text{data}}(\boldsymbol{x}_0)d\boldsymbol{x}_0, \tag{29}$$

and construct the following kernel:

$$\kappa_t(\boldsymbol{x}, \boldsymbol{x}') = \frac{p_t(\boldsymbol{x}, \boldsymbol{x}')}{p_t(\boldsymbol{x})p_t(\boldsymbol{x}')}. \tag{30}$$

Equivalently, we can write

$$\kappa_t(\boldsymbol{x}, \boldsymbol{x}') = \int \frac{q_t(\boldsymbol{x} \mid \boldsymbol{x}_0)}{p_t(\boldsymbol{x})}\frac{q_t(\boldsymbol{x}' \mid \boldsymbol{x}_0)}{p_t(\boldsymbol{x}')}p_{\text{data}}(\boldsymbol{x}_0)d\boldsymbol{x}_0. \tag{31}$$

We first show that $\kappa_t$ is a valid symmetric positive semidefinite kernel. The symmetry directly follows from

$$p_t(\boldsymbol{x}, \boldsymbol{x}') = \int q_t(\boldsymbol{x} \mid \boldsymbol{x}_0)q_t(\boldsymbol{x}' \mid \boldsymbol{x}_0)p_{\text{data}}(\boldsymbol{x}_0)d\boldsymbol{x}_0 = p_t(\boldsymbol{x}', \boldsymbol{x}). \tag{32}$$

Next, for any finite set of points $\{\boldsymbol{x}_i\}_{i=1}^n$ and coefficients $\{c_i\}_{i=1}^n$, we have

$$\sum_{i,j=1}^n c_i c_j \kappa_t(\boldsymbol{x}_i, \boldsymbol{x}_j) = \sum_{i,j=1}^n c_i c_j \int \frac{q_t(\boldsymbol{x}_i \mid \boldsymbol{x}_0)}{p_t(\boldsymbol{x}_i)}\frac{q_t(\boldsymbol{x}_j \mid \boldsymbol{x}_0)}{p_t(\boldsymbol{x}_j)}p_{\text{data}}(\boldsymbol{x}_0)d\boldsymbol{x}_0 \tag{33}$$

$$= \int \left[\sum_{i=1}^n c_i \frac{q_t(\boldsymbol{x}_i \mid \boldsymbol{x}_0)}{p_t(\boldsymbol{x}_i)}\right]^2 p_{\text{data}}(\boldsymbol{x}_0)d\boldsymbol{x}_0 \tag{34}$$

$$\geq 0. \tag{35}$$

Thus, $\kappa_t$ is positive semidefinite.

For the Gaussian perturbation kernel used in diffusion models,

$$q_t(\boldsymbol{x}_t \mid \boldsymbol{x}_0) = \mathcal{N}\left(\boldsymbol{x}_t; s(t)\boldsymbol{x}_0, s(t)^2\sigma(t)^2\boldsymbol{I}\right), \tag{36}$$

if $|s(t)|\sigma(t) > 0$, then $q_t(\boldsymbol{x}_t \mid \boldsymbol{x}_0) > 0$ for all $\boldsymbol{x}_t$ and $\boldsymbol{x}_0$. Hence,

$$p_t(\boldsymbol{x}) = \int q_t(\boldsymbol{x} \mid \boldsymbol{x}_0)p_{\text{data}}(\boldsymbol{x}_0)d\boldsymbol{x}_0 > 0 \tag{37}$$

Therefore, $\kappa_t$ is symmetric and positive semidefinite.

We further assume:

$$\iint \kappa_t^2(\boldsymbol{x}, \boldsymbol{x}')p_t(\boldsymbol{x})p_t(\boldsymbol{x}')d\boldsymbol{x}d\boldsymbol{x}' < \infty. \tag{38}$$

Under this condition, the associated kernel integral operator is a compact, self-adjoint, and positive operator on $L^2(p_t)$.

$$(\mathcal{K}_t h)(\boldsymbol{x}) = \int \kappa_t(\boldsymbol{x}, \boldsymbol{x}')h(\boldsymbol{x}')p_t(\boldsymbol{x}')d\boldsymbol{x}'. \tag{39}$$

Therefore, it admits an orthonormal eigendecomposition $\{(\mu_{t,l}, \psi_{t,l})\}_{l=0}^{\infty}$ satisfying

$$\mathcal{K}_t\psi_{t,l} = \mu_{t,l}\psi_{t,l}, \tag{40}$$

and

$$\int \psi_{t,l}(\boldsymbol{x})\psi_{t,m}(\boldsymbol{x})p_t(\boldsymbol{x})d\boldsymbol{x} = \delta_{lm}. \tag{41}$$

Moreover, the kernel admits the spectral expansion

$$\kappa_t(\boldsymbol{x}, \boldsymbol{x}') = \sum_{l=0}^{\infty} \mu_{t,l}\psi_{t,l}(\boldsymbol{x})\psi_{t,l}(\boldsymbol{x}'). \tag{42}$$

Now we prove the diffusion-distance expansion. By definition,

$$D_{\kappa_t}^2(\boldsymbol{x}, \boldsymbol{x}') = \int \left[\kappa_t(\boldsymbol{x}, \boldsymbol{y}) - \kappa_t(\boldsymbol{x}', \boldsymbol{y})\right]^2 p_t(\boldsymbol{y})d\boldsymbol{y}. \tag{43}$$

Using the spectral expansion of $\kappa_t$, we have

$$\kappa_t(\boldsymbol{x}, \boldsymbol{y}) - \kappa_t(\boldsymbol{x}', \boldsymbol{y}) = \sum_{l=0}^{\infty} \mu_{t,l}\psi_{t,l}(\boldsymbol{x})\psi_{t,l}(\boldsymbol{y}) - \sum_{l=0}^{\infty} \mu_{t,l}\psi_{t,l}(\boldsymbol{x}')\psi_{t,l}(\boldsymbol{y}) \tag{44}$$

$$= \sum_{l=0}^{\infty} \mu_{t,l}\left[\psi_{t,l}(\boldsymbol{x}) - \psi_{t,l}(\boldsymbol{x}')\right]\psi_{t,l}(\boldsymbol{y}). \tag{45}$$

Substituting this into Equation 43, we obtain

$$D_{\kappa_t}^2(\boldsymbol{x}, \boldsymbol{x}') = \int \left[\sum_{l=0}^{\infty} \mu_{t,l}\left[\psi_{t,l}(\boldsymbol{x}) - \psi_{t,l}(\boldsymbol{x}')\right]\psi_{t,l}(\boldsymbol{y})\right]^2 p_t(\boldsymbol{y})d\boldsymbol{y} \tag{46}$$

$$= \int \sum_{l,m=0}^{\infty} \mu_{t,l}\mu_{t,m}\left[\psi_{t,l}(\boldsymbol{x}) - \psi_{t,l}(\boldsymbol{x}')\right]\left[\psi_{t,m}(\boldsymbol{x}) - \psi_{t,m}(\boldsymbol{x}')\right]\psi_{t,l}(\boldsymbol{y})\psi_{t,m}(\boldsymbol{y})p_t(\boldsymbol{y})d\boldsymbol{y} \tag{47}$$

$$= \sum_{l,m=0}^{\infty} \mu_{t,l}\mu_{t,m}\left[\psi_{t,l}(\boldsymbol{x}) - \psi_{t,l}(\boldsymbol{x}')\right]\left[\psi_{t,m}(\boldsymbol{x}) - \psi_{t,m}(\boldsymbol{x}')\right]\int \psi_{t,l}(\boldsymbol{y})\psi_{t,m}(\boldsymbol{y})p_t(\boldsymbol{y})d\boldsymbol{y} \tag{48}$$

$$= \sum_{l,m=0}^{\infty} \mu_{t,l}\mu_{t,m}\left[\psi_{t,l}(\boldsymbol{x}) - \psi_{t,l}(\boldsymbol{x}')\right]\left[\psi_{t,m}(\boldsymbol{x}) - \psi_{t,m}(\boldsymbol{x}')\right]\delta_{lm} \tag{49}$$

$$= \sum_{l=0}^{\infty} \mu_{t,l}^2\left[\psi_{t,l}(\boldsymbol{x}) - \psi_{t,l}(\boldsymbol{x}')\right]^2. \tag{50}$$

Therefore, the diffusion distance admits the eigenspace expansion

$$D^2_{\kappa_t}(\boldsymbol{x}, \boldsymbol{x}') = \sum_{l=0}^{\infty} \mu^2_{t,l} \left[ \psi_{t,l}(\boldsymbol{x}) - \psi_{t,l}(\boldsymbol{x}') \right]^2 . \tag{51}$$

## C. Duality of Spectral Representation Learning and Closed-form Diffusion Score Distillation

We adopt the result of Garrido et al. (2022) that dimension-contrastive and sample-contrastive self-supervised objectives are equivalent when representation embeddings are normalized across channels and mini-batches. The spectral regularization can finally have this equivalent form:

$$\min_{\theta} - \sum_{i=1}^{B} \psi_{\theta}(\boldsymbol{x}_i, t)^{\top} \psi_{\theta}(\boldsymbol{x}'_i, t) + \sum_{i=1}^{B} \sum_{j \neq i} \psi_{\theta}(\boldsymbol{x}_i, t)^{\top} \psi_{\theta}(\boldsymbol{x}_j, t) \tag{52}$$

$$\Leftrightarrow \min_{\theta} - \sum_{i=1}^{B} \left( \frac{\psi_{\theta}(\boldsymbol{x}_i, t)^{\top} \psi_{\theta}(\boldsymbol{x}'_i, t)}{\tau} \right) + \sum_{i=1}^{B} \log \left[ \sum_{j \neq i} \exp \left( \frac{\psi_{\theta}(\boldsymbol{x}_i, t)^{\top} \psi_{\theta}(\boldsymbol{x}_j, t)}{\tau} \right) \right], \tag{53}$$

where $\tau$ denotes a temperature hyperparameter. As the spectral embedding $\psi(\boldsymbol{x}_i, t)$ is normalized, the above optimization problem can be further re-written as the following one:

$$\min_{\theta} - \underbrace{\sum_{i=1}^{B} \log \left[ \exp \left( \frac{-\|\psi_{\theta}(\boldsymbol{x}_i, t) - \psi_{\theta}(\boldsymbol{x}'_i, t)\|_2^2}{\tau} \right) \right]}_{:= \mathcal{L}_s^+} \tag{54}$$

$$+ \underbrace{\sum_{i=1}^{B} \log \left[ \sum_{j \neq i} \exp \left( \frac{-\|\psi_{\theta}(\boldsymbol{x}_i, t) - \psi_{\theta}(\boldsymbol{x}_j, t)\|_2^2}{\tau} \right) \right]}_{:= \mathcal{L}_s^-}, \tag{55}$$

where we transform the dot product operations to L2 distance. Interestingly, when $\psi_{\theta}(\boldsymbol{x}_j, t)$ in $\mathcal{L}_s^-$ and $\psi_{\theta}(\boldsymbol{x}'_i, t)$ in $\mathcal{L}_s^+$ are detached from gradient propagation (which is true in our adopt NeuralEF (Deng et al., 2022b) approach), their derivatives regarding $\psi_{\theta}(\boldsymbol{x}_i, t)$ are in the similar form of batch-wise closed-form score of diffusion models in the representation embedding space:

$$\nabla_{\psi_{\theta}(\boldsymbol{x}_i, t)} \mathcal{L}_s^+ = \frac{2}{\tau} \left( \psi_{\theta}(\boldsymbol{x}_i, t) - \psi_{\theta}(\boldsymbol{x}'_i, t) \right) \tag{56}$$

$$\nabla_{\psi_{\theta}(\boldsymbol{x}_i, t)} \mathcal{L}_s^- = \frac{2}{\tau} \sum_{k \neq i} \frac{\exp \left( -\|\psi_{\theta}(\boldsymbol{x}_i, t) - \psi_{\theta}(\boldsymbol{x}_k, t)\|_2^2 / \tau \right)}{\sum_{j \neq i} \exp \left( -\|\psi_{\theta}(\boldsymbol{x}_i, t) - \psi_{\theta}(\boldsymbol{x}_j, t)\|_2^2 / \tau \right)} \left( \psi_{\theta}(\boldsymbol{x}_k, t) - \psi_{\theta}(\boldsymbol{x}_i, t) \right) \tag{57}$$

The gradient expressions in Equation 57 and 56 resemble the closed-form score of diffusion models (Scarvelis et al., 2023). Given a training set $\mathcal{D} = \{\boldsymbol{x}_i\}_{i=0}^{D}$ with $D$ samples, the closed-form expression of the score function under the rectified flow formulation can be written as:

$$\nabla_{\boldsymbol{z}} \log p_t(\boldsymbol{z}) = \frac{1}{t^2} \sum_{k=1}^{D} \frac{\exp \left( -\|\boldsymbol{z} - (1-t)\boldsymbol{x}_k\|_2^2 / 2t^2 \right)}{\sum_{j=1}^{D} \exp \left( -\|\boldsymbol{z} - (1-t)\boldsymbol{x}_j\|_2^2 / 2t^2 \right)} \left( (1-t)\boldsymbol{x}_k - \boldsymbol{z} \right), \tag{58}$$

where $\boldsymbol{z} = (1-t)\boldsymbol{x} + t\boldsymbol{\epsilon}$, $\boldsymbol{x} \sim \mathcal{D}$, $\boldsymbol{\epsilon} \sim \mathcal{N}(0, I)$, $\forall t \in (0, 1]$. By comparing equations 58 and 57: the temperature $\tau$ can be seen as $2t^2$, the counterparts of $\psi_{\theta}(\boldsymbol{x}_k, t)$ in the numerator and $\psi_{\theta}(\boldsymbol{x}_j, t)$ in the denominator are $(1-t)\boldsymbol{x}_k$ and $(1-t)\boldsymbol{x}_j$, and data samples for evaluating the gradient in Equation 57 are those negative samples. The notation in Equation 56 is defined analogously; the difference is that the score is evaluated at a single positive sample.

In this sense, the total derivative $\partial \mathcal{L}_s / \partial \psi_{\theta}(\boldsymbol{x}_i, t) = \nabla_{\psi_{\theta}(\boldsymbol{x}_i, t)} \mathcal{L}_s^+ + \nabla_{\psi_{\theta}(\boldsymbol{x}_i, t)} \mathcal{L}_s^-$ is a score function evaluated on a sampled data batch. Intuitively, $\nabla_{\psi_{\theta}(\boldsymbol{x}_i, t)} \mathcal{L}_s^-$ points at the direction which is a weighted sum of displacement vectors from $\psi_{\theta}(\boldsymbol{x}_i, t)$

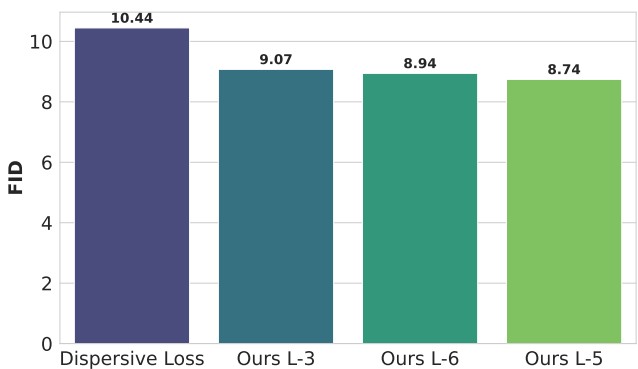

*Figure 5.* We compare an alternative dispersive loss with our spectral loss on CIFAR10 dataset. In addition, an ablation study is conducted to assess the impact of the layer choice for spectral alignment.

to $\psi_\theta(\boldsymbol{x}_k, t)$ for all $k \neq i, k \in [B]$. The pairwise weights decrease with the squared L2 distances and are normalized by the softmax function. Once $\psi_\theta$ is learned to represent eigenfunctions, the displacement vectors are weighted by the diffusion distance (without eigenvalue weighting) of data samples (see Appendix B). Conversely, $\nabla_{\psi_\theta(\boldsymbol{x}_i,t)}\mathcal{L}_s^+$ points away from the positive sample's representation $\psi_\theta(\boldsymbol{x}_i', t)$, akin to the negative-prompting in diffusion models.

Next, we can show that optimizing our spectral regularization term is actually conducting a score distillation. For $\boldsymbol{x} \sim p_t(\boldsymbol{x})$, $\psi_\theta(\cdot, t)$ can be seen as a generator: $\psi_\theta(\boldsymbol{x}, t) \sim p_t^{\psi_\theta}$, where $p_t^{\psi_\theta}$ is a latent distribution of spectral embeddings. A score distillation step from $p_t^{\psi_\theta}$ to a target distribution $p_{\text{target}}$ can be achieved by minimizing their KL divergence through a gradient-based optimizer. Specifically, the gradient of KL divergence w.r.t $\theta$ is:

$$\nabla_\theta D_{\text{KL}}(p_t^{\psi_\theta} \parallel p_{\text{target}}) = \mathbb{E}_{\boldsymbol{x} \sim p_t}\left[ (\nabla_\theta \psi_\theta(\boldsymbol{x}, t))^\top \left( \nabla_{\psi_\theta(\boldsymbol{x},t)} \log p_t^{\psi_\theta} - \nabla_{\psi_\theta(\boldsymbol{x},t)} \log p_{\text{target}} \right) \right] \tag{59}$$

Let $p_{\text{target}}$ be a Gaussian mixture centered at positive samples with bandwidth $\tau$ (in our case, there is only one positive sample), and model the latent distribution $p_t^{\psi_\theta}$ as a Gaussian mixture over negative samples with the same bandwidth $\tau$, we have $\nabla_{\psi_\theta(\boldsymbol{x},t)} \log p_t^{\psi_\theta} = \nabla_{\psi_\theta(\boldsymbol{x}_i,t)}\mathcal{L}_s^-$ and $\nabla_{\psi_\theta(\boldsymbol{x},t)} \log p_{\text{target}} = -\nabla_{\psi_\theta(\boldsymbol{x}_i,t)}\mathcal{L}_s^+$.

Therefore, the gradient of the score distillation step turns out to be:

$$\nabla_\theta D_{\text{KL}}(p_t^{\psi_\theta} \parallel p_{\text{target}}) = \mathbb{E}_{\boldsymbol{x} \sim p_t}\left[ (\nabla_\theta \psi_\theta(\boldsymbol{x}, t))^\top \left( \nabla_{\psi_\theta(\boldsymbol{x},t)}\mathcal{L}_s^- + \nabla_{\psi_\theta(\boldsymbol{x},t)}\mathcal{L}_s^+ \right) \right] \tag{60}$$

$$= \mathbb{E}_{\boldsymbol{x} \sim p_t}\left[ (\nabla_\theta \psi_\theta(\boldsymbol{x}, t))^\top \nabla_{\psi_\theta(\boldsymbol{x},t)}\mathcal{L}_s \right] \tag{61}$$

By the chain rule, the gradient of the original spectral representation objective w.r.t $\theta$ is:

$$\frac{\partial \mathcal{L}_s}{\partial \theta} = \mathbb{E}_{\boldsymbol{x} \sim p_t}\left[ (\nabla_\theta \psi_\theta(\boldsymbol{x}, t))^\top \nabla_{\psi_\theta(\boldsymbol{x},t)}\mathcal{L}_s \right] \equiv \nabla_\theta D_{\text{KL}}(p_t^{\psi_\theta} \parallel p_{\text{target}}) \tag{62}$$

This concludes the proof that shows optimizing the spectral representation regularizer is performing diffusion score distillation.

## D. Additional Experiment Details and Results

### D.1. Synthetic Distributions

In Figure 7, we compare our approach with the baseline on four additional 2D patterns. Overall, our method captures the geometric structure of each distribution more tightly at earlier training stages, whereas the baseline samples remain more dispersed and noisy. Figure 6 visualizes hidden states probed from our model and the baseline. In the high-noise regime ($t = 0.5$ and $0.7$), the two methods produce broadly similar hidden representations. In contrast, in the low-noise regime ($t = 0.97$ and $0.98$), our method yields cleaner and more clearly separated representations. This provides further evidence that the proposed spectral regularization helps organize the internal feature space, thereby improving the model's ability to fit the geometric structure of 2D patterns.

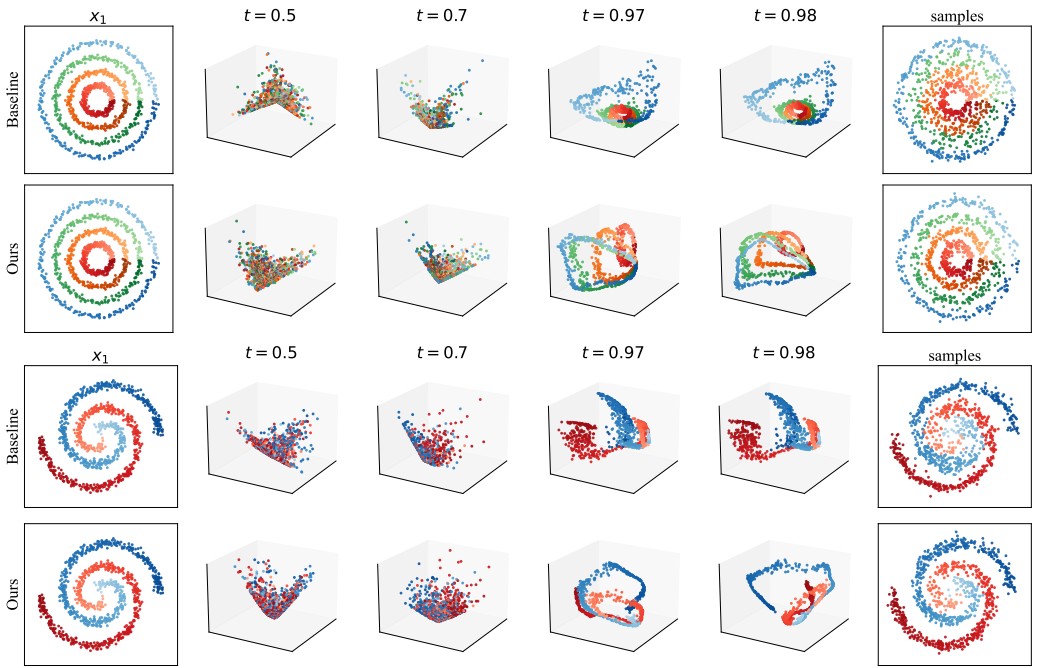

*Figure 6.* Visualization of model hidden states on synthetic 2D patterns using PCA. We project the hidden states of both our method and the baseline to three principal components. The visualized hidden states are probed from the third layer of the MLP networks.

### D.2. Image Generation

**Training details.** We use DiT (Peebles & Xie, 2023) as the base model and employ the parameterization and training objective of rectified flow (Liu et al., 2022). For small datasets (CIFAR-10, CelebA, FFHQ), to mitigate overfitting, we train a small DiT (S, 13M parameters) and patchify images into $2 \times 2$ pixel patches (patch size 2). For ImageNet $64 \times 64$ experiment (models work in pixel space), we train an L/4 model (558M parameters, patch size 4). For ImageNet $256 \times 256$ experiment (models work in latent space), we follow the XL/2 configuration of Peebles & Xie (2023), yielding a 681M-parameter model. Training schedules are adjusted to the dataset scales: S/2 models on CIFAR-10, CelebA, and FFHQ are trained and evaluated at 70K iterations; ImageNet $64 \times 64$ models are trained and evaluated at 100K iterations; and the latent ImageNet $256 \times 256$ model is trained and evaluated at 400K iterations. Since our spectral regularizer requires an additional batch of perturbed samples, we halve the base batch size so that each optimizer step processes the same total number of training examples.

**More results.** In Figure 5, we present comparison results on CIFAR10 dataset between our model with varying choices of alignment layer and dispersive loss (Wang & He, 2025) (an alternative self-supervised alignment approach).

### D.3. Protein/RNA Generation on Manifold

We extend our evaluation to protein and RNA generation, adhering to the experimental protocols established by Chen & Lipman (2023); Huang et al. (2022). We adopt the torsion angle datasets (Lovell et al., 2003; Murray et al., 2003) curated by Huang et al. (2022). Our spectral alignment is integrated into a Riemannian flow matching framework defined on 2D and 7D torus. As shown in Table 3, the spectral-aligned model consistently exceeds the performance of the vanilla baseline. In particular, the performance margin widens in the 7D case, suggesting that spectral alignment is particularly effective at regularizing flows on this type of more complex high-dimensional manifold.

*Table 3.* Test NLL on protein & RNA datasets.

|  | General (2D) | Glycine (2D) | Proline (2D) | Pre-Pro (2D) | RNA (7D) |
|---|---|---|---|---|---|
| Riemannian FM | $1.022_{\pm 0.021}$ | $1.947_{\pm 0.023}$ | $0.169_{\pm 0.024}$ | $1.196_{\pm 0.032}$ | $-4.780_{\pm 0.196}$ |
| Ours | $\mathbf{1.018}_{\pm 0.028}$ | $\mathbf{1.935}_{\pm 0.014}$ | $\mathbf{0.161}_{\pm 0.029}$ | $\mathbf{1.192}_{\pm 0.039}$ | $\mathbf{-5.167}_{\pm 0.083}$ |

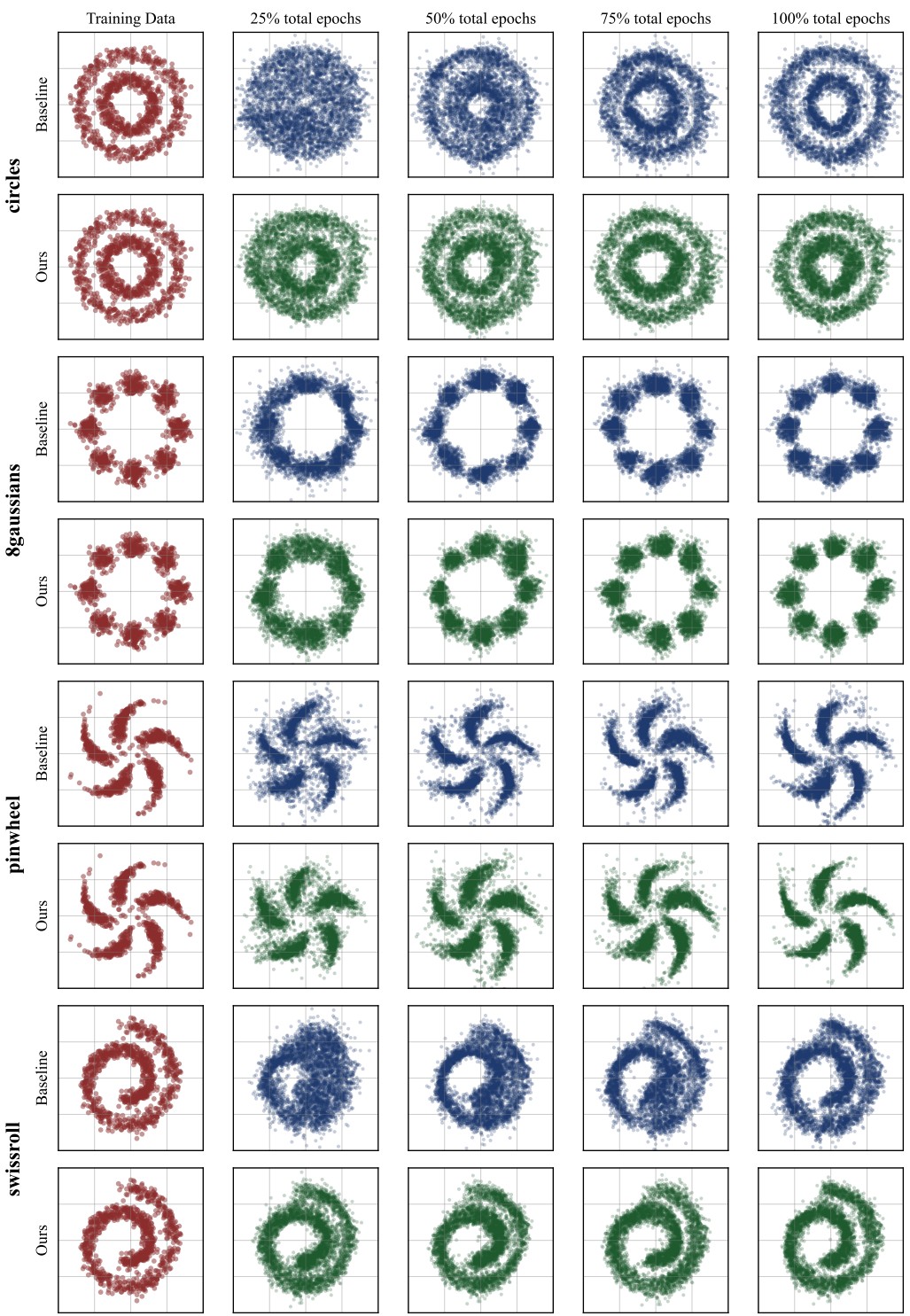

*Figure 7.* Training progress of the baseline diffusion model and our method on four 2D point distributions. We visualize samples from checkpoints at 25%, 50%, 75%, and 100% of training progress on circles, 8-gaussians, pinwheel, and swissroll distributions.

