# OpenReview forum: "Revisiting Spectral Representations in Generative Diffusion Models"
_ICML.cc/2026/Conference — ICML 2026 regular_

### Official Review · Reviewer_gfvt · 2026-03-11

**Soundness:** 2
**Presentation:** 3
**Significance:** 2
**Originality:** 3
**Overall Recommendation:** 4
**Confidence:** 4

**Summary:**

The paper proposes a method that uses self-supervised spectral representation learning (SRL) to improve the training of the diffusion models. As SRL and diffusion models both learn from perturbed data, the authors propose the spectral representation alignment loss based on the Gaussian perturbation kernel of the diffusion model. The diffusion denoiser and the representation model share the same backbone to enable efficient training. Experiments on various image generation datasets and point cloud generation show the empirical performance of the proposed methods.

**Compliance With Llm Reviewing Policy:**

Affirmed.

**Final Justification:**

My concerns have been adequately addressed in the authors' rebuttal. I think the weak acceptance rating is appropriate for the paper.

**Key Questions For Authors:**

- Could the authors explain the reasons behind employing the ODE sampler for pixel-space generation while utilizing the SDE sampler for latent-space generation?
- Does the inclusion of the additional SRL loss term introduce significant computational overhead during training?

**Limitations:**

The authors do not discuss the limitations of the work. I suggest the authors to add a limitation section in the appendix.

**Strengths And Weaknesses:**

### Strength
- The paper is clearly written, and the proposed method is well-motivated.
- The proposed method is easy to implement and applicable to various architectures.

### Weakness
- Limited empirical gains compared to prior work: The primary concern is that the performance improvement of the proposed method appears modest relative to REPA. As shown in Table 1, the FID scores remain notably behind current state-of-the-art approaches. For instance, on CIFAR-10, the results are substantially worse than those of the strong baseline EDM [1]. To better substantiate the effectiveness of the proposed method, I recommend including comparisons with more competitive baselines under comparable experimental settings.

[1]. Elucidating the Design Space of Diffusion-Based Generative Models, NeurIPS 2022.

-  Insufficient analysis of classifier-free guidance (CFG): The authors fix CFG=2 across all experiments in Table 1. However, it is standard practice to report results both with and without CFG, and to explore performance under the optimal CFG scale and scheduling interval. Providing a more comprehensive ablation study on CFG hyperparameters would help clarify the method's empirical robustness and practical utility.

---

> ### Author Rebuttal · Authors · 2026-03-31
>
> We appreciate the reviewer's efforts in reviewing our submission.
>
> **1. Performance comparison on EDM**
>
> We appreciate the reviewers' concern regarding the choice of baseline architecture. Our decision to use DiT rather than the EDM U-Net is deliberate and primarily motivated by the scope and positioning of this work. DiT has become a standard backbone for modern diffusion and rectified-flow models due to its strong scalability and broad applicability across modalities. In addition, our method is closely related in spirit to REPA, which is also built on a DiT backbone, making DiT the most natural architecture for isolating the effect of our proposed spectral regularization.
> Adapting the same representation-alignment framework to a U-Net is not a trivial drop-in change. In particular, U-Nets do not have a single uniform hidden-token interface as in DiT: one must decide which hidden states to probe across encoder and decoder stages with residual skip connections, and one must also address the fact that intermediate U-Net hidden states are high-dimensional spatial feature maps. Applying representation alignment thereby requires additional architectural choices.
> Our goal on image-generation tasks is to evaluate the effectiveness of spectral regularization itself, rather than to exhaustively optimize architecture-specific design choices or build the strongest possible image-generation system. For this reason, we chose a DiT baseline that is better aligned with our problem setting and with prior representation-alignment work. We also note that it has been discussed in [a DiT codebase issue](https://github.com/facebookresearch/DiT/issues/84) that DiT does not achieve particularly strong FID on small-scale datasets such as CIFAR-10 (same in Fig 3 (100k) of UViT paper ), and thus results around FID ~= 10 are within the expected range for this baseline.
>
> To further address the reviewers' concern, we additionally conducted a supplementary experiment using the EDM codebase on CIFAR-10 to verify that the benefit of our method is not specific to DiT. In this experiment, we probe hidden states from the downsampling encoder of the U-Net, flatten them, project them to a 4096-dimensional embedding space using an MLP (16384 -> 8192 -> 4096), and then apply our spectral regularization in that embedding space. As shown below, our method still improves both FID and IS under the EDM codebase, indicating that the benefit of spectral regularization is not tied to a particular backbone architecture.
> | | FID | IS |
>  | :--- | :--- | :--- |
> | Baseline (DDPM++) | 2.457 | 9.491 |
>  | Ours (backbone: DDPM++) | 2.381 | 9.523 |
>
> **2. Choice of ODE and SDE samplers**
>
> Our sampler choice follows the standard settings of the corresponding experimental setup and codebase. The main goal is to minimize additional confounding factors and avoid introducing extra hyperparameter tuning that is orthogonal to the contribution of this paper. For pixel-space generation, we use the simplest ODE sampler, which introduces no additional hyperparameters beyond the number of sampling steps and CFG. For latent-space generation, we follow the REPA codebase, where the default setting uses an SDE sampler. To ensure a fair comparison, we keep the sampler configuration unchanged and follow their default setup in our experiments.
>
> **3. Analysis of CFG**
>
> We use standard CFG values for generation and do not tune CFG to optimize the performance of our method. This choice keeps the comparison fair and avoids introducing additional task-specific tuning that is orthogonal to the main point of the paper. While some works include a CFG ablation, such studies (like DiT, SD, meanflow) are more commonly emphasized when the primary contribution lies in the probabilistic modeling or large image generation. In our case, the contribution is a general training improvement, and thus we focus the main experiments on comparing methods under standard generation settings.
> As an additional reference, we provide a CFG ablation on ImageNet-64 in the FID table below. Across a representative range of commonly used CFG values for DiT, our method consistently outperforms the baseline. These results suggest that the improvement is robust to the choice of CFG.
> | | CFG=1.0 | CFG=2.0 | CFG=3.0 |
>  | :--- | :--- | :--- | :--- |
> | Baseline | 9.389 | 9.441 | 16.8412 |
> | Ours | 9.095 | 7.994 | 14.352 |
>
> **4. Computational overhead during training**
> The training overhead of our method depends on the backbone size, as it is primarily determined by the dimension of the intermediate representations used for regularization. We measure this overhead in three representative settings: ImageNet-64 for pixel-space, ImageNet-256 for latent-space generation, and Airplane in shapenet for point-cloud generation. Compared to the corresponding baseline, our method increases training time by approximately 10% on pixel-space generation, 70% on latent-space generation, and 20% on Airplane point-cloud generation.

---

> > ### Author Rebuttal · Reviewer_gfvt · 2026-04-06
> >
> > I thank the authors for the detailed rebuttal. As my concerns have been adequately addressed, I will update the rating accordingly.

---

### Official Review · Reviewer_QH7w · 2026-03-12

**Soundness:** 3
**Presentation:** 3
**Significance:** 2
**Originality:** 3
**Overall Recommendation:** 4
**Confidence:** 4

**Summary:**

This paper proposes a method that regularizes the intermediate representations of diffusion models with self-supervised spectral representation learning to improve generation performance. In the proposed method, the intermediate representations are aligned with the eigenfunctions of a time-dependent kernel integral operator corresponding to the diffusion process, encouraging the model to learn representations that preserve the local geometry of the data manifold. Furthermore, the authors provide an analysis showing that the gradient of this spectral regularization is equivalent to score distillation in the representation space, which gives the method a theoretical grounding. Experimentally, the method demonstrates consistent improvements on image generation and 3D point cloud generation datasets. A notable advantage is that it does not depend on external pretrained encoders.

**Compliance With Llm Reviewing Policy:**

Affirmed.

**Final Justification:**

Most of my concerns have been addressed by the authors' rebuttal. However, I still have some remaining concerns regarding the ablation study. On the other hand, I appreciate the important advantage of the proposed method in that it does not rely on external encoders, and I have no particular objection on this point. Additionally, the paper as a whole is well-organized, and the writing is generally clear.

Furthermore, I felt that the scope of the claims needed to be appropriately qualified, but this point has been sufficiently clarified through the rebuttal.

Taking all of this into account, I believe it is reasonable to judge this paper as a Weak Accept, and I am raising my score accordingly.

**Key Questions For Authors:**

The baseline FID values on CIFAR-10, CelebA, and FFHQ are considerably worse than those reported by existing strong diffusion models, and on latent ImageNet 256, which is closer to a stronger setting, the proposed method underperforms REPA. Given these results, the observed improvements may reflect a phenomenon where auxiliary regularization is generally more effective when applied to insufficiently trained baselines. In light of this concern, can you conclude that the effect specific to spectral representation learning has been sufficiently demonstrated?

**Limitations:**

yes

**Strengths And Weaknesses:**

### Strengths

- One of the key strengths of this work is that the representation regularization does not depend on external pretrained encoders. While existing methods such as REPA achieve performance gains by using powerful pretrained representations (e.g., DINOv2) as teachers, the proposed method derives spectral representations from the perturbation structure inherent in the diffusion process itself. This eliminates the need for large-scale pretrained teachers exposed to additional data, and makes the method applicable to domains where suitable external encoders are not readily available, such as 3D point clouds. This broad applicability is a clear advantage.

- The paper starts from the problem of how to improve representation learning in diffusion models, and naturally motivates the proposed method by identifying the limitations of existing approaches. In particular, the connection between diffusion processes and spectral representation learning through the shared perspective of perturbation kernels is presented in a coherent and well-motivated manner. While this is a subjective assessment, I found the overall structure of the argument easy to follow and the paper generally well-written.

### Weaknesses

- The experimental results require careful interpretation regarding the effectiveness of the proposed method. In particular, the baseline generation performance on CIFAR-10, CelebA, and FFHQ is not strong in absolute terms and falls considerably short of the results reported by existing strong diffusion models. As a result, it is difficult to disentangle whether the observed improvements are due to the proposed spectral regularization specifically, or whether auxiliary representation regularization is generally more effective in undertrained or underpowered settings. Indeed, on latent ImageNet 256, which is closer to a strong comparison setting, the improvement is limited and the method underperforms REPA. Therefore, it is difficult to conclude from the current results that the proposed method is generally effective when applied to strong baselines. At a minimum, without comparisons against more sufficiently trained baselines, compute-matched comparisons, and detailed comparisons with existing strong representation regularization methods, the validity of the claims remains limited.

- The ablation studies are insufficient for isolating the essential components of the proposed method. In particular, experiments examining the sensitivity to the regularization coefficient $\lambda$, the necessity of the additional perturbed sample $x'_t$, and comparisons of the spectral loss itself against other simple auxiliary losses are lacking. As a result, it is difficult to determine whether the improvements are specifically attributable to the proposed spectral representation learning or whether they could be explained by a general regularization effect. This makes the causal support for the claims somewhat weak.

- The 2D experiments are primarily qualitative and lack quantitative evaluation. While the visualization of 3K samples shows that the geometric structure is better captured, no quantitative metrics are provided to support the claims made on the 2D distributions. Although these experiments serve as a natural illustration for the geometric prior argument, they do not constitute strong empirical evidence.

- The description of experimental results contains somewhat overgeneralized claims. The paper states that performance gains are consistent, but in Table 1, on ImageNet 64, while FID improves, IS decreases and Precision also slightly degrades. Therefore, the expression "consistent gains" is not strictly accurate with respect to all metrics, and the paper should specify which evaluation metrics show improvement.

### Minor Comments

- Around Equation (5), the score function is parameterized by $\phi$, but the objective is written as a function of $\theta$.
- Immediately after Equation (11), "$f \in L^2(\mathcal{X}, p)$" should presumably be "$h \in L^2(\mathcal{X}, p)$".
- In the Qualitative Results paragraph, the text states that Figure 4 shows comparisons for the "Car" and "Airplane" categories, but the figure appears to show the "Airplane" and "Chair" categories instead.

---

> ### Author Rebuttal · Authors · 2026-03-31
>
> We thank the reviewer for their detailed review of our submission.
>
> **1. Concerns on strong baseline performance**
>
> We appreciate this concern and agree that the empirical claims should be interpreted carefully.
> Our intention is not to claim that the current DiT-based instantiation outperforms the strongest absolute diffusion baselines on CIFAR-10, CelebA, FFHQ, or latent ImageNet-256. Rather, our claim is that spectral regularization provides a consistent improvement over a matched diffusion backbone, and that this improvement is not restricted to a particular architecture or to the very early stage of training.
>
> First, regarding the relatively weak absolute performance: this is primarily attributable to the choice of backbone. We adopt DiT as the main architecture because it is the standard backbone in recent diffusion work and is also the natural setting for comparison to the REPA-style approach. However, DiT is generally less competitive than strong U-Net-based models such as EDM/DDPM++ on small-scale image datasets. For example, as shown in Figure 3 of [1], when a DiT-style model (U-ViT) is trained on CIFAR-10 for **100K** iterations (compared to **70K** in our setting), *its FID typically remains in the range of 11-15*. To directly address the reviewer's concern, we additionally evaluated our method on a strong U-Net baseline (EDM) on CIFAR-10. Even in this stronger setting, our method still improves both FID and IS. **The strong baseline results and experiment settings are given in the reply to R4**.
>
> Second, regarding whether the gain is limited to undertrained settings: Figure 3 shows that our method begins to outperform the baseline after a certain number of iterations, and that the gap is then maintained throughout the remainder of training. This behavior suggests a persistent improvement instead of only helping in an undertrained regime.
>
> Third, we agree that REPA is a strong and highly relevant baseline. In our preliminary experiments, REPA on low-resolution images is not effective (see **1. in the reply to R2**). Our goal in this paper is not to optimize for the strongest possible performance in such large-scale, highly tuned settings, but to test a more fundamental question: whether a principled self-supervised spectral regularization can improve diffusion training in a general way. Further gains in scalable latent-generation regimes typically require substantially more compute, as well as careful system-level and domain-specific engineering, which lie beyond the scope of this work.
>
> We will revise the paper to make this scope explicit and avoid overgeneralization.
>
> [1] Bao, F., et al. (2023). All are worth words: A vit backbone for diffusion models.
>
> **2. Ablation studies on $\lambda$**
>
> We conducted an ablation study on the weight $\lambda$ on CIFAR-10, and report the results in the table below. The method is stable across a reasonable range of values, with $\lambda \in \{0.05, 0.5, 0.1\}$ giving consistently good performance. Since the goal of our experiments is to validate the effectiveness of the proposed principled regularization, rather than to exhaustively tune for the best possible or state-of-the-art result, we do not perform extensive hyperparameter optimization. Based on this ablation, we found $\lambda = 0.05$ to be a simple and effective choice, and use it consistently in all image-generation experiments.
> | $\lambda$| 0.05 | 0.5 | 0.1 |
> | :--- | :--- | :--- | :--- |
> | FID | 8.742 | 8.578 | 8.628 |
>
> **3. Ablation studies on additional perturbed samples**
>
> Without the additional perturbed samples, the self-supervised objective is no longer uniquely specified, and there are many possible alternatives. One representative choice is the dispersive loss [2], which we compare against in Appendix Figure 5 on CIFAR-10. That variant underperforms our method by **~1.7 FID**, indicating that the additional perturbed samples are a necessary part of the proposed design.
>
> [2] Wang, R., et al. (2025). Diffuse and disperse: Image generation with representation regularization
>
> **4. Quantitative evaluation of 2D experiments**
>
> To quantitatively evaluate whether our approach improves over the baseline, we use the widely adopted Wasserstein distance (WD) to measure the discrepancy between the generated 2D distribution and the true data distribution. The results are reported in the table below. On the 2spirals pattern, our method reduces the Wasserstein distance by **12.4%** relative to the baseline. On the pinwheel pattern, the distribution generated by our method is **48.8%** closer to the true distribution. These are consistent with the qualitative results.
>
> | Method | 2spirals WD | pinwheel WD |
> | :--- | :--- | :--- |
> | Baseline | 0.0179 | 0.0298 |
> | Ours | 0.0157 | 0.0153 |
>
> **5. Overgeneralized claims & typos**
>
> We appreciate the reviewer's suggestion and will avoid overclaims in experiments and those typos and minor inconsistencies.

---

> > ### Author Rebuttal · Reviewer_QH7w · 2026-04-02
> >
> > Thank you for the rebuttal. Most of my concerns have been addressed.
> >
> > However, I still have some reservation regarding point 3, “Ablation studies on additional perturbed samples.” The comparison against dispersive loss is useful and suggests that the gain is not explained by an arbitrary auxiliary regularizer. However, I am still not fully convinced that the improvement is specific to the proposed spectral objective itself, rather than a broader class of auxiliary feature regularization methods.
> >
> > Could the authors clarify why the comparison against dispersive loss is sufficient to support this point? In particular, why should dispersive loss be considered a representative alternative when arguing that the benefit comes from the proposed spectral objective itself?

---

> > > ### Author Response · Authors · 2026-04-06
> > >
> > > We thank the reviewer for the follow-up. The comparison with dispersive loss is intended to address the specific ablation study of whether the additional perturbed sample contributes meaningfully to performance. For this purpose, dispersive loss is a natural reference: it is a simple self-supervised regularizer for diffusion models that operates in a similar spirit, but does not use the additional perturbed samples in our formulation. In this sense, it provides a relevant comparison for isolating the benefit of this component.
> > >
> > > More broadly, we agree that a wider class of auxiliary representation regularizers may also facilitate diffusion training. Our claims do *not* include that the proposed spectral regularization is the unique or universally best such regularizer. Rather, the goal of this paper is to show that the basic and canonical spectral regularization arising from the perturbation-kernel perspective yields an effective and principled self-supervised representation alignment objective for diffusion models without external encoders, and that this objective works well in practice.
> > >
> > > The proposed perspective through diffusion distance and the duality to diffusion score distillation may also provide a basis for developing more advanced auxiliary self-supervised alignment objectives for diffusion models. In this paper, we focus on establishing the effectiveness of the canonical spectral formulation itself.

---

### Official Review · Reviewer_cwze · 2026-03-12

**Soundness:** 3
**Presentation:** 3
**Significance:** 3
**Originality:** 3
**Overall Recommendation:** 4
**Confidence:** 4

**Summary:**

This paper investigates the connection between diffusion-based generative models and self-supervised spectral representation learning through a joint perspective of perturbation kernels. Under this joint perspective, diffusion models generate by reversing the noise-injection processes specified by Gaussian kernels; spectral embeddings emerge from contrasting positive and negative pairs induced by random perturbation kernels. Motivated by the joint perspective, the authors extended REPA to facilitate training of diffusion-based generative models, and demonstrate how the training can be improved from a geometric perspective. They also draw a connection between the optimization of the proposed objective and diffusion score distillation in the representation space. They have also shown empirical effectiveness on synthetic data, image data (CIFAR10, CelebA, FFHQ, ImageNet) and point cloud data (ShapeNet).

**Compliance With Llm Reviewing Policy:**

Affirmed.

**Final Justification:**

I appreciate the authors' detailed and thoughtful rebuttal, which has satisfactorily addressed my questions. I have no further concerns at this point. I will keep my current score, as my initial rating was intentionally set on the higher side and already reflected my view that the post-rebuttal paper is in the borderline accept range.

**Key Questions For Authors:**

1. See weaknesses.
2. The authors mentioned that the joint spectral learning can benefit diffusion training from a geometric perspective. I would like to ask the authors to highlight the evidences by referring to them in the paper. Also, would it be helpful to visualize the latent representations to demonstrate how the internal geometry is better organized?
3. I found some interesting results in the Appendix (comparison to dispersive loss and performance on protein/RNA datasets in Appendix D). Would it be helpful to at least briefly mention them in the main text?

**Limitations:**

Yes

**Strengths And Weaknesses:**

Strengths
1. The abstract is nicely written. The motivation and connection to existing works are well discussed. It is fairly easy to follow the narrative and the flow of the story.
2. The proposed method is simple and elegant. Figure 1 represents the method in a very concise manner.
3. The empirical advantage over the baseline diffusion transformer (DiT) is consistent and clear. I particularly like Figure 3 that highlights training stage where the performance of the proposed method starts to take the lead. Figure 4 also shows visually obvious improvement over DiT-3D in point cloud generation.

Weaknesses
1. Since this is almost a direct improvement on REPA, I believe it would be somewhat necessary to have some direct competitions with REPA. So far, I only see Table 1 contains one experiment (ImageNet, res = 256, latent) and it seems to be performing not as good. I would expect the authors to provide some additional results at least in the image generation task, and, in case not outperforming REPA, provide some reasonable justifications.

---

> ### Author Rebuttal · Authors · 2026-03-31
>
> We appreciate the reviewer's efforts in providing insightful comments.
>
> **1. REPA on pixel-space generation**
>
> We reimplemented REPA for our pixel-space image generation experiments, as the original method is designed for latent diffusion models. On CIFAR-10, however, REPA with a DINOv2 ViT-B/14 teacher does not yield competitive optimization or generation performance. After 50K iterations, it reaches an FID of 37.84, which is substantially worse than both the baseline (11.26) and our method (9.38) under the same training budget.
>
> We believe this is likely due to a mismatch between the pretrained DINOv2 representation space and the intermediate features needed for diffusion training on low-resolution images. This result also underscores an important distinction between the two approaches: REPA depends on an external pretrained teacher whose representation quality may not transfer well across data scales and domains, whereas our method is self-supervised and adapts naturally to the target distribution.
>
> **2. Latent representation visualization**
>
> We agree that visualization would be helpful for illustrating the claimed geometric effect. In the revision, we will include representation visualizations for the 2D experiments, where the learned geometry can be shown most directly. Specifically, we will project the learned embeddings to 2D with PCA to better illustrate the structural differences induced by the proposed regularization.
> For image and point-cloud experiments, comparable visualization is less informative, since the probed representations are high-dimensional latent features without a direct correspondence to image patches or individual points. Consequently, simple low-dimensional projections in these settings are harder to interpret. We therefore believe the 2D experiments provide the clearest setting for visualizing the geometric connection.
>
> **3. Results in the appendix**
>
> We thank the reviewer for the suggestions, and will mention our appendix results of comparison to dispersive loss and protein/RNA generation in the main text.

---

> > ### Author Rebuttal · Reviewer_cwze · 2026-04-01
> >
> > I am satisfied with the response and have no further questions.

---

### Official Review · Reviewer_dFQ2 · 2026-03-13

**Soundness:** 3
**Presentation:** 3
**Significance:** 3
**Originality:** 3
**Overall Recommendation:** 4
**Confidence:** 4

**Summary:**

- This paper unifies diffusion models and spectral representation learning through the lens of perturbation kernels.
- It introduces a self-supervised spectral regularization that aligns intermediate representations with eigenfunctions of the kernel,

**Compliance With Llm Reviewing Policy:**

Affirmed.

**Final Justification:**

After reviewing the rebuttal, I still maintain my original judgment.

**Key Questions For Authors:**

- I am curious whether this spectral regularization approach can be extended to graph generation?
- Do you think your spectral regularization framework could be naturally extended to such bridge models, where the perturbation kernel is no longer Gaussian but defined by a bridge process?

**Limitations:**

- The duality proof in Proposition 4.3 and Appendix C relies on assumptions about embedding normalization and batch-wise sampling that may not hold perfectly during training

**Strengths And Weaknesses:**

### Strengths
- Achieves improved diffusion model training with relatively low computational overhead by incorporating spectral information (e.g., eigenfunctions) through a lightweight projection head, avoiding expensive matrix decomposition.
- Operates entirely in the neural network feature space without requiring explicit numerical decomposition (such as eigen-decomposition or FFT), making it scalable and easy to integrate into existing diffusion training pipelines.
### Weaknesses
- The duality proof in Proposition 4.3 and Appendix C relies on assumptions about embedding normalization and batch-wise sampling that may not hold perfectly during training

---

> ### Author Rebuttal · Authors · 2026-03-31
>
> We thank the reviewer for the constructive comments.
>
> **1. Assumptions about embedding normalization and batch-wise sampling**
>
> The assumptions used in our duality analysis follow the results in [1] and are commonly adopted for theoretical tractability. In implementation, we enforce normalization using standard PyTorch operators, whose numerical approximation error is negligible in practice.
>
> [1] Garrido, Q., Chen, Y., Bardes, A., Najman, L., & Lecun, Y. (2022). On the duality between contrastive and non-contrastive self-supervised learning.
>
> **2. Spectral regularization extended to graph generation**
>
> For many diffusion-based graph generation problems, including molecular generation, the diffusion model is typically defined on node states, while edges are recovered or predicted from the generated node properties. In this sense, such tasks are closely related to point-cloud generation, where the primary challenge is also to model structured relationships through node-level geometry. This is one reason we consider our point-cloud results to be representative evidence for broader geometric-structure generation settings.
> To directly examine applicability to graphs, we further conducted a simple experiment based on the recent graph autoencoder and graph diffusion framework of [2] on the Ego-small graph generation task. We trained both the baseline and our method for 5k iterations, and adopted the same MMD metrics used in [2] to compare graph-statistic distributions. As shown in the table below, our method reduces the average metric by 25.5%, suggesting that the benefit of spectral regularization extends beyond point clouds to graph generation as well.
>
> [2] Wen, L., Tang, X., Ouyang, M., Shen, X., Yang, J., Zhu, D., ... & Wei, X. (2024). Hyperbolic graph diffusion model.
>
> | Metric | Baseline | Ours | Change |
> | :--- | :--- | :--- | :--- |
> | degree | 0.2618 | **0.2299** | -12.2% |
> | cluster | 0.5759 | **0.3919** | -32.0% |
> | orbit | 0.00872 | **0.00855** | -2.0% |
> | **mean** | 0.2821 | 0.2101 | **-25.5%** |
>
> **3. Spectral regularization framework extended to non-Gaussian perturbation kernel**
>
> As a practical regularization method, our approach can potentially be applied to diffusion/bridge models with non-Gaussian perturbation kernels. However, the present theoretical justification for the duality does not directly carry over. This is because the score-distillation argument relies on the closed-form diffusion score under Gaussian perturbations, and such a form is generally unavailable for non-Gaussian kernels.

---

> > ### Author Rebuttal · Reviewer_dFQ2 · 2026-04-04
> >
> > After reviewing the rebuttal, I still maintain my original judgment.

---

### Decision · Program_Chairs · 2026-04-30

**Decision:**

Accept (regular)

**Comment:**

This paper is well-motivated, with a simple and elegant solution. There were some concerns around the baselines being considerably worse than those of existing diffusion models and REPA performing better in several cases. The narrative, presentation and the idea make this paper interesting to the ICML community.